# A Linearly Convergent GAN Inversion-based Algorithm for Reverse Engineering of Deceptions

## Abstract

An important aspect of developing reliable deep learning systems is devising strategies that make these systems robust to adversarial attacks. There is a long line of work that focuses on developing defenses against these attacks, but recently, researchers have begun to study ways to *reverse engineer the attack process*. This allows us to not only defend against several attack models, but also classify the threat model. However, there is still a lack of theoretical guarantees for the reverse engineering process. Current approaches that give any guarantees are based on the assumption that the data lies in a union of linear subspaces, which is not a valid assumption for more complex datasets. In this paper, we propose a novel framework for reverse engineering of deceptions which supposes that the clean data lies in the range of a GAN. To classify the signal and attack, we jointly solve a GAN inversion problem and a block-sparse recovery problem. The core contribution of this paper is to provide for the first time deterministic *linear convergence guarantees* for this problem. We also empirically demonstrate the merits of the proposed approach on several nonlinear datasets as compared to state-of-the-art methods.

## 1 Introduction

Modern deep neural network classifiers have been shown to be vulnerable to imperceptible perturbations to the input that can drastically affect the prediction of the classifier. These adversarially attacked inputs can pose problems in safety-critical applications where correct classification is paramount. Adversarial attacks can be either fixed universal perturbations, which can deceive a pretrained network on different images of the same dataset (Moosavi-Dezfooli et al., 2017), or image-dependent perturbations (Poursaeed et al., 2018). For the latter approach, attack generation for a given classification network entails maximizing a classification loss function subject to various constraints (Madry et al., 2018). For instance, we can assume that the additive perturbation $\delta$ for a clean signal $x$ lies in an $\ell_p$ ball for some $p \geq 1$, i.e., $\delta \in \mathcal{S}_p$, where $\mathcal{S}_p = \{\delta : \|\delta\|_p \leq 1\}$ (Maini et al., 2020).

Over the last few years, there has been significant interest in the topic of devising defenses to enhance the adversarial robustness of deep learning systems. This constant endeavor has led to a growing interest in methods that adopt a more holistic approach towards adversarial robustness, known as the *Reverse Engineering of Deceptions (RED)* problem. The objective of RED is to go beyond mere defenses by simultaneously *defending against the attack* and *inferring the deception strategy* followed to corrupt the input, e.g., which $\ell_p$ norm was used to generate the attack (Gong et al., 2022). There are various practical methods to reverse engineer adversarial attacks that rely on deep representations of corrupted signals or complicated ad-hoc architectures Moayeri & Feizi (2021); Gong et al. (2022); Goebel et al. (2021) , but their effectiveness is only empirically verified, and there is a noticeable lack of theoretical guarantees for the RED problem.

This inspired the work of Thaker et al. (2022), in which the authors propose the first principled approach for the RED problem. Specifically, for additive $\ell_p$ attacks, they assume that both *the signal $x$ and the attack $\delta$ live in unions of linear subspaces* spanned by the blocks of dictionaries $D_s$ and $D_a$ that correspond to the signal and the attack respectively i.e. $x = D_s c_s$ and $\delta = D_a c_a$. These dictionaries are divided into blocks according to the classes of interest for $x$ and $\delta$ (i.e., the signal classification labels for $x$ and the type of $\ell_p$ threat model used for generating $\delta$). The specific form of

$D_s$ and $D_a$ gives rise to block-sparse representations for the signal $x$ and the attack $\delta$ with respect to these dictionaries. This motivates their formulation of RED as an inverse optimization problem where the representation vectors $c_s$ and $c_a$ of the clean signal $x$ and attack $\delta$ are estimated under a block-sparse promoting regularizer, i.e.,

$$\min_{c_s,c_a} \|x' - \underbrace{D_s c_s}_{x} - \underbrace{D_a c_a}_{\delta}\|_2 + \lambda_s \|c_s\|_{1,2} + \lambda_a \|c_a\|_{1,2}. \tag{1}$$

Above, $\|\cdot\|_{1,2}$ is a block-sparsity promoting $\ell_1/\ell_2$ norm (Stojnic et al., 2009; Eldar & Mishali, 2009; Elhamifar & Vidal, 2012). To solve this problem, Thaker et al. (2022) use an alternating minimization algorithm for estimating $c_s$ and $c_a$ and accordingly provide theoretical recovery guarantees for the correctness of their approach.

While these recent works undoubtedly demonstrate the importance of the problem of reverse engineering of deceptions (RED), there still exist several challenges.

**Challenges.** While the approach in Thaker et al. (2022) is theoretically sound, the main drawback of the method is that it is comes with strong assumptions on the data generative model, i.e. that the data live in a union of linear subspaces. It is apparent that this assumption is unrealistic for complex and high-dimensional datasets. A natural step towards relaxing this simplistic assumption is to *leverage the power of deep generative models*, thus building on adversarial purification approaches and suitably adjusting their formulation to the RED problem. However, we are then left with an inverse problem that is highly non-convex. Namely, the signal reconstruction involves a projection step onto the manifold parameterized by a pretrained deep generative model, and this problem is yet to be theoretically understood. Further, RED involves finding *latent representations for both the signal and the attack*. An efficient way to deal with this is to use an *alternating minimization algorithm*, as in Thaker et al. (2022). This leads to the following challenge for developing both practical and theoretically grounded algorithms:

> *Can we provide theoretical guarantees for an alternating minimization algorithm that minimizes a non-convex and non-smooth RED objective?*

**Contributions.** In this work, we propose *a novel reverse engineering of deceptions approach* that offers *theoretical guarantees* and can be applied to *complex datasets*. We address the weakness of the work in Thaker et al. (2022) by leveraging the power of nonlinear deep generative models. Specifically, we replace the signal model $x = D_s c_s$ in equation 1 with $x = G(z)$, where $G : \mathbb{R}^d \to \mathbb{R}^n, d \ll n$ is the generator of a Generative Adversarial Network (GAN). By using a pre-trained GAN generator, we can reconstruct the clean signal by projecting onto the signal manifold learned by the GAN, i.e. by estimating a $z$ such that $G(z) \approx x$. Further, adversarial perturbations are modeled as in Thaker et al. (2022), i.e. as block-sparse vectors with respect to a predefined dictionary. The inverse problem we solve in this model is then:

$$\min_{z,c_a} \|x' - \underbrace{G(z)}_{x} - \underbrace{D_a c_a}_{\delta}\|_2 + \lambda \|c_a\|_{1,2}. \tag{2}$$

Our main contributions are the following:

- *A GAN inversion-based RED algorithm with linear convergence guarantees.* We address the main challenge above and provide recovery guarantees for the RED problem. We divide our analysis into two regimes.

  1. The first setting is the clean signal reconstruction problem. For this problem, known as GAN inversion, we demonstrate linear convergence of a subgradient descent algorithm to the global minimizer of the objective function. Note that we only require *smoothness* of the activation function and a *local error-bound* condition. To the best of our knowledge, this is the *first result that analyzes the GAN inversion problem departing from the standard assumption of networks with randomized weights* (Hand & Voroninski, 2017; Joshi et al., 2021).

  2. We then add adversarial noise and solve Equation 2 to give linear convergence guarantees of an alternating descent algorithm that alternates between updating $z$ and $c_a$. This allows us to significantly extend the work of Thaker et al. (2022) while maintaining theoretical guarantees.

- *Empirical Results for the RED problem.* We also empirically verify our theoretical results on simulated data and demonstrate strong performance on the RED problem using our alternating algorithm on the MNIST, Fashion-MNIST and CIFAR-10 datasets.

## 2 RELATED WORK

**Adversarial Defenses.** We restrict our discussion of adversarial attacks (Carlini & Wagner, 2017; Biggio et al., 2013), to the *white-box attack* scenario where adversaries have access to the network parameters and solve a loss maximization problem. Adversarial training, a min-max optimization approach, has been the most popular defense strategy Tramer & Boneh (2019). Adversarial purification methods rely on pretrained deep generative models as a prior for denoising corrupted images (Nie et al., 2022; Samangouei et al., 2018). This is formulated as an inverse optimization problem (Xia et al., 2022), whose theoretical properties are still not fully understood (Joshi et al., 2021; Hand & Voroninski, 2017). In contrast to the wide defenses literature , our work focuses on the theoretical aspects of the RED problem and the corresponding inverse problems.

**Theoretical Analysis of GAN inversion algorithms.** We employ a GAN inversion strategy for the RED problem. There is a rich history of deep generative models for inverse problems, such as compressed sensing (Ongie et al., 2020; Jalal et al., 2020) super-resolution (Menon et al., 2020), image inpainting (Xia et al., 2022). However, theoretical understanding of the GAN inversion optimization landscape has studied settings where the GAN has random or close-to-random weights (Shah & Hegde, 2018; Hand & Voroninski, 2017; Joshi et al., 2021; Lei et al., 2019; Song et al., 2019). For the first time in the literature, we depart from these assumptions to provide a more holistic analysis of the GAN inversion problem, instead leveraging error-bound conditions and proximal Polyak-Łojasiewicz conditions (Karimi et al., 2016; Frei & Gu, 2021; Drusvyatskiy & Lewis, 2018).

**Reverse Engineering of Deceptions (RED).** RED is a recent framework to not only defend against attacks, but also reverse engineer and infer the type of attack. In Goebel et al. (2021), the authors train a multi-class network to detect a corrupted image and predict attack attributes. In Gong et al. (2022), a denoiser learns by aligning the predictions of the denoised input and the clean signal. Moayeri & Feizi (2021) use pretrained self-supervised embeddings e.g. SimCLR (Chen et al., 2020) to classify the attacks in the low-data regime. In contrast, our work is focused on the theoretical aspects of the RED problem; as such, we focus on improving upon the work of Thaker et al. (2022), in which the authors show a provably correct block-sparse optimization approach for RED. The main drawback of Thaker et al. (2022) is that their modelling assumption for the clean signal is often violated in practice, which motivates our modelling assumptions.

## 3 PROBLEM FORMULATION

We build on Thaker et al. (2022) to develop a model for an adversarial example $x' = x + \delta$, with $x$ being the clean signal and $\delta$ the adversarial perturbation. We replace the model of equation 1 with a pretrained generator $G$ of a GAN. Thus, the generative model we assume for $x$ is given by

$$x' \approx G(z) + D_a c_a. \tag{3}$$

We use generators $G : \mathbb{R}^d \to \mathbb{R}^{n_L}, d \gg n_L$ which are $L$-layer networks of the form

$$G(z) = \sigma(W_L \sigma(W_{L-1} \cdots W_2 \sigma(W_1 z))) \tag{4}$$

where $W_i \in \mathbb{R}^{n_i \times n_{i-1}}$ are the known GAN parameters with $n_0 = d$, $\sigma$ is a nonlinear activation function, and $D_a \in \mathbb{R}^{n_L \times k_a}$ is an attack dictionary (typically with $k_a > n_L$).

As in Thaker et al. (2022), the attack dictionary $D_a$ contains blocks corresponding to different $\ell_p$ attacks (for varying $p$) computed on training samples of each class. Thaker et al. (2022) verify this modelling assumption by showing that for networks that use piecewise linear activations, $\ell_p$ attacks evaluated on test examples can be expressed as linear combinations of $\ell_p$ attacks evaluated on training examples. Using the model in equation 3, we then formulate an inverse problem to learn $z$ and $c_a$:

$$\min_{z, c_a} \mathcal{L}(z, c_a) \triangleq f(z, c_a) + \lambda h(c_a), \tag{5}$$

where $f(z, c_a) = \|x' - G(z) - D_a c_a\|_2^2$ denotes a reconstruction loss and $h(c_a)$ denotes a (nonsmooth) convex regularizer on the coefficients $c_a$. For example, in Thaker et al. (2022), the regularizer $h(c_a)$ is $\|c_a\|_{1,2}$ which promotes block-sparsity on $c_a$ according to the structure of $D_a$. We note that our theoretical results do not assume this form for $D_a$, but rather that its spectrum is bounded.

A natural algorithm to learn both $z$ and $c_a$ is to alternate between updating $z$ via subgradient descent and $c_a$ via proximal gradient descent, as shown in Algorithm 1.

---

**Algorithm 1** Proposed RED Algorithm

---

Given: $x' \in \mathbb{R}^{n_L}, G : \mathbb{R}^d \to \mathbb{R}^{n_L}, D_a \in \mathbb{R}^{n_L \times k_a}$
Initialize: $z^0, c_a^0$
Set: Step size $\eta$ and regularization parameter $\lambda$
    **for** $k = 0, 1, 2, \ldots$ **do**
        $R_i \leftarrow \mathrm{diag}(\sigma'(W_i z^k))$ for $i \in \{1, \ldots, L\}$
        $z^{k+1} \leftarrow z^k - \eta (W_1 R_1)^T (W_2 R_2)^T \cdots (W_L R_L)^T (G(z^k) + D_a c_a^k - x')$
        $c_a^{k+1} \leftarrow \mathrm{prox}_{\lambda h} \left\{ c_a^k - \eta D_a^T (G(z^k) + D_a c_a^k - x') \right\}$
    **end for**
    **return** $z^{k+1}, c_a^{k+1}$

---

## 4    MAIN RESULTS: THEORETICAL GUARANTEES FOR RED

In this section, we provide our main theoretical results for the RED problem by demonstrating the convergence of the iterates of Algorithm 1 to global optima. A priori, this is difficult due to the non-convexity of equation 5 introduced by the GAN generator $G(z)$ (Hand & Voroninski, 2017). To get around this issue, works such as Hand & Voroninski (2017) and Huang et al. (2021) make certain assumptions to avoid spurious stationary points. However, these conditions essentially reduce to the GAN having weights that behave as a random network (see Definition 12 in Appendix). In practice, especially for the RED problem, modelling real data often requires GANs with far-from-random weights, so there is a strong need for theoretical results in this setting.

We draw inspiration from the theory of deep learning and optimization literature, where several works have analyzed non-convex problems through the lens of Polyak-Łojasiewicz (PL) conditions or assumptions that lead to benign optimization landscapes (Karimi et al., 2016; Richards & Kuzborskij, 2021; Liu et al., 2022). Our goal is to depart from the randomized analysis of previous GAN inversion works to address the non-convexity of the problem. The main assumption we employ is a *local error bound* condition. We conjecture this assumption holds true in practice for two reasons. First, we show that the random network conditions assumed in existing works (Hand & Voroninski, 2017; Huang et al., 2021) already imply a local error bound condition (see Corollary 4). Moreover, in Section 5.1, we give examples of non-random networks that also empirically satisfy the local error-bound condition, showing the generality of our assumption. Secondly, the empirical success of GAN inversion in various applications suggests that the optimization landscape is benign (Xia et al., 2022). However, for the GAN inversion problem, traditional landscape properties such as a PL condition do not hold globally [1]. Nevertheless, we can use local properties of benign regions of the landscape to analyze convergence[2]. Our work serves as an initial step to analyze convergence of far-from-random networks, and an important avenue of future work is verifying the local error bound condition theoretically for certain classes of networks.

We divide our analysis into three settings, eventually arriving at our main goal which is to provide convergence results for the RED problem. First, we study the problem with no adversarial noise i.e. $x' = x + \delta$ where $\delta = 0$. This is known as the GAN inversion problem, and Section 4.1 provides linear convergence guarantees for GAN inversion, along with comparison to existing theoretical results in Section 4.1.1. Next, in Section 4.2, we add adversarial noise and provide convergence guarantees when using alternating gradient descent to optimize Equation 5 in the unregularized case i.e. $\lambda = 0$. Finally, in Section 4.3, we study the regularized case of Equation 5.

---

[1] We refer the reader to Section 3 of (Liu et al., 2022) for a simple explanation of this phenomenon.
[2] Note that similar local conditions to analyze convergence have been used in works analyzing the theory of deep learning, such as (Liu et al., 2022).

## 4.1 Convergence Analysis of the GAN Inversion Problem

To begin with the most simplified version of the problem, we begin with the GAN inversion problem, which is the case when there is no adversarial noise added to the signal. This simply corresponds to finding the latent code $z$ for an input $x$ and fixed GAN $G$ such that $G(z) = x$. We let $f(z) \triangleq \|x - G(z)\|_2^2$. Suppose there exists a $z^*$ such that $x' = G(z^*)$, so $z^*$ is a global minimizer of $f(z)$. Our first set of results will ensure convergence of the iterates $z^k$ to $z^*$. We will denote $\|\Delta z^{k+1}\|_2 \triangleq \|z^{k+1} - z^*\|_2$. Since the GAN inversion problem is highly non-convex, in order to prove our convergence results, we need to posit some assumptions on $G$ and the algorithm iterates.

**Assumption 1.** *(Activation Function) We assume that $\sigma$ is twice differentiable and smooth.*

Note that standard activation functions such as the sigmoid or tanh or smooth ReLU variants (softplus, GeLU, Swish etc.) satisfy Assumption 1.

**Assumption 2.** *(Local Error Bound Condition) For all $z^k$ on the optimization trajectory, suppose that there exists a $\mu > 0$ such that*

$$\left\|\nabla_z f(z^k)\right\|_2^2 \geq \mu^2 \left\|\Delta z^k\right\|_2^2 \tag{6}$$

Under these assumptions, our first main result demonstrates linear convergence of the iterates $z^k$ to the global minimizer $z^*$ of the GAN inversion problem.

**Theorem 3.** *Suppose that Assumption 1 holds for the nonlinear activation function and Assumption 2 holds with local error bound parameter $\mu$. Let $\rho$ and $-\epsilon$ be the maximum and minimum eigenvalues of the Hessian of the loss. Further, assume that the step size satisfies $\eta \leq \min\left\{\frac{1}{4\epsilon}, \frac{3}{2\rho}\right\}$ and $\eta \in \left(\frac{3\mu^2 - \sqrt{9\mu^4 - 32\mu^2\rho\epsilon}}{4\mu^2\rho}, \frac{3\mu^2 + \sqrt{9\mu^4 - 32\mu^2\rho\epsilon}}{4\mu^2\rho}\right)$. Lastly, assume that $\mu \gtrsim \sqrt{\rho\epsilon}$. Then, we have that the iterates converge linearly to the global optimum with the following rate in $(0, 1)$:*

$$\left\|\Delta z^{k+1}\right\|_2^2 \leq \left(1 - 4\eta^2\mu^2\left(\frac{3}{4} - \frac{\eta\rho}{2}\right) + 4\eta\epsilon\right)\left\|\Delta z^k\right\|_2^2 \tag{7}$$

The proof is deferred to the Appendix. Inspired by the proof strategy of Richards & Kuzborskij (2021), we show an almost co-coercivity of the gradient (Lemma 9 in Appendix) that depends on bounding $\rho$ and $\epsilon$ for smooth and twice differentiable activation functions.

Along with the step size $\eta$, there are three problem-specific parameters that affect the convergence rate: the largest and the smallest eigenvalues of the Hessian of the loss, i.e., $\rho$ and $-\epsilon$ respectively, and the local error bound parameter $\mu$. Note that because the problem is non-convex, the Hessian will have at least one negative eigenvalue. The rate becomes closer to 1 and convergence slows as $\epsilon$ gets larger because $\epsilon$ controls the slack in co-coercivity of the gradient in our proof. Similarly, if the operator norm of the weights is controlled, then the convergence rate is faster as a function of $\rho$. Finally, the convergence rate speeds up as $\mu$ increases since each gradient descent iterate takes a larger step towards the minimizer. The condition $\mu \gtrsim \sqrt{\rho\epsilon}$ ensures that the gradient norm is roughly larger than the negative curvature of the Hessian, so that progress towards the global minimizer can still be maintained. The quantity $\sqrt{\rho\epsilon}$ is the geometric mean of the largest and smallest eigenvalue of the Hessian and can be thought of as a quantity capturing the range of the spectrum of the Hessian.

Due to the almost co-coercivity property of the gradient operator (see Lemma 9, Appendix), the step size of gradient descent must be bounded away from zero. However, for practical purposes, the regime that is useful for ensuring fast convergence is when the step size is indeed sufficiently large.

### 4.1.1 Comparison to Existing GAN Inversion Approaches

Hand & Voroninski (2017) and Huang et al. (2021) derive a condition on the GAN weights, known as Weight Distribution Condition (WDC), under which they characterize the optimization landscape of the GAN inversion problem. The WDC ensures the weights of the network behave close to random networks (see Definition 12 in Appendix). Hand & Voroninski (2017) show that under the WDC,

there is only one spurious stationary point with a small basin of attraction. We provide a different viewpoint by demonstrating that the WDC implies a local error bound condition with parameter $\mu$.

**Corollary 4.** *(GAN Inversion for Networks that satisfy WDC) Let $\epsilon$ be fixed such that $K_1 L^8 \epsilon^{1/4} \leq 1$, where $L$ is the number of the layers of the GAN generator and $K_1$ an absolute constant. Suppose that for all $i \in [L]$, $W_i$ satisfies the WDC with parameter $\epsilon$. Suppose we initialize the iterates $z^0$ of Algorithm 1 that satisfy*

$$z^0 \notin \mathcal{B}(z^*, K_2 L^3 \epsilon^{1/4} \left\| z^* \right\|_2) \cup \mathcal{B}(-\kappa z^*, K_2 L^{13} \epsilon^{1/4} \left\| z^* \right\|_2) \cup \{0\} \tag{8}$$

*where $\mathcal{B}(c, r)$ denotes an $\ell_2$ ball with center $c$ and radius $r$, $K_2$ denotes an absolute constant and $\kappa \in (0, 1)$. Then, there exists $\mu > 0$ such that the local error bound condition holds along the optimization trajectory of subgradient descent applied to iterate $z^0$.*

To illustrate the generality of the local error bound condition, we show in Section 5.1 that the local error bound condition can also hold for certain classes of non-random networks.

## 4.2 Reverse Engineering of Deceptions Optimization Problem without Regularization

Next, we generalize the results of the previous section to the unregularized RED setting where in Algorithm 1, we only minimize $f(z, c_a)$, i.e. $\lambda = 0$ and $\text{prox}_{\lambda h}(\cdot)$ is the identity function. Suppose there exists a $z^*$ and $c_a^*$ such that $x' = G(z^*) + D_a c_a^*$, so $(z^*, c_a^*)$ are global minimizers of $f(z, c_a)$. Our first set of results will ensure convergence of the iterates $(z^k, c_a^k)$ to $(z^*, c_a^*)$. We will denote $\left\| \Delta z^{k+1} \right\|_2 \triangleq \left\| z^{k+1} - z^* \right\|_2$ and $\left\| \Delta c_a^{k+1} \right\|_2 \triangleq \left\| c_a^{k+1} - c_a^* \right\|_2$. In order to capture the behaviour of $c_a^k$ on the optimization landscape, we refine Assumption 2 to a different local error bound condition.

**Assumption 5.** *(Local Error Bound Condition) For all $z^k$ and $c_a^k$ on the optimization trajectory, suppose that there exists a $\mu > 0$ such that*

$$\left\| \nabla_z f(z^k, c_a^k) \right\|_2^2 + \left\| \nabla_{c_a} f(z^k, c_a^k) \right\|_2^2 \geq \mu^2 (\left\| \Delta z^k \right\|_2^2 + \left\| \Delta c_a^k \right\|_2^2) \tag{9}$$

Under these assumptions, our main theorem for the RED problem demonstrates linear convergence of the iterates $z^k$ and $c_a^k$ to the global minimizers $z^*$ and $c_a^*$.

**Theorem 6.** *Suppose that Assumption 1 holds for the nonlinear activation function and Assumption 5 holds with local error bound parameter $\mu$. Let $\rho$ and $-\epsilon$ be the maximum and minimum eigenvalues of the Hessian of the loss. Under the same assumption on the step size $\eta$ and the local error bound parameter $\mu$ as in Theorem 3, we have that the iterates converge linearly to the global optimum with the following rate in $(0, 1)$:*

$$\left\| \Delta z^{k+1} \right\|_2^2 + \left\| \Delta c_a^{k+1} \right\|_2^2 \leq \left( 1 - 4\eta^2 \mu^2 \left( \frac{3}{4} - \frac{\eta\rho}{2} \right) + 4\eta\epsilon \right) (\left\| \Delta z^k \right\|_2^2 + \left\| \Delta c_a^k \right\|_2^2) \tag{10}$$

The proof is also deferred to the Appendix.

## 4.3 Regularized Reverse Engineering of Deceptions Optimization Problem

We now consider the regularized problem, with $\lambda \neq 0$. The analysis presented in Section 4.2 does not immediately extend to this setting because $(z^*, c_a^*)$ now denote minimizers of $\mathcal{L}(z, c_a) = f(z, c_a) + \lambda h(c_a)$, which is not necessarily the pair $(z^*, c_a^*)$ such that $x' = G(z^*) + D_a c_a^*$. In order to demonstrate convergence, we appeal to well-known results that use the Polyak-Łojasiewicz (PL) condition. We assume a local proximal PL condition on the iterates $c_a^k$, which can be thought of as a version of Assumption 5 but on the function values instead of the iterates (Karimi et al., 2016). This assumption also takes into account the proximal update step for $c_a$ [3].

**Assumption 7.** *Let $\rho$ denote the Lipschitz constant of the gradient of $f$ with respect to both $z$ and $c_a$. For all $z^k$ and $c_a^k$ on the optimization trajectory, suppose that there exists a $\mu > 0$ such that*

$$2\rho \mathcal{D}(c_a^k, \rho) + \left\| \nabla_z f(z^k, c_a^k) \right\|_2^2 \geq \mu(\mathcal{L}(z^k, c_a^k) - \mathcal{L}(z^*, c_a^*)) \tag{11}$$

---

[3]We refer the reader to Karimi et al. (2016) for intuition on the global proximal PL inequality

where $\mathcal{D}(c_a^k, \rho) = -\min_y \left[ \langle \nabla_{c_a} f(z^k, c_a^k), y - c_a^k \rangle + \frac{\rho}{2} \| y - c_a^k \|_2^2 + h(y) - h(c_a^k) \right]$

**Theorem 8.** *Suppose Assumption 7 holds with constant $\mu > 0$. Let $\rho$ be the maximum eigenvalue of the Hessian of the loss. If $h$ is convex and $\eta = \frac{1}{\rho}$, then the function values converge linearly:*

$$\mathcal{L}(z^{k+1}, c_a^{k+1}) - \mathcal{L}(z^*, c_a^*) \leq \left( 1 - \frac{\mu}{2\rho} \right) (\mathcal{L}(z^k, c_a^k) - \mathcal{L}(z^*, c_a^*)) \tag{12}$$

The proof is in the Appendix, but we note similarities to Karimi et al. (2016), Theorem 5.

## 5 EXPERIMENTS

In this section, we provide experiments to verify the local error bound condition, as well as demonstrate the success of our approach on the MNIST, Fashion-MNIST, and CIFAR-10 datasets.

### 5.1 VERIFICATION OF THE LOCAL ERROR BOUND CONDITION

By studying a realizable RED problem instance, we will demonstrate that the local error bound condition holds for a variety of random and non-random GANs. First, we set up a binary classification task on data $x$ generated from a one-layer GAN $G(z) = \sigma(Wz)$ with $W \in \mathbb{R}^{m \times d}$. For a fixed classification network $\psi(x)$, we generate adversarial attacks. Since our problem is realizable, we can compute the error bound parameter $\mu$ exactly. The full experimental setup is given in the Appendix.

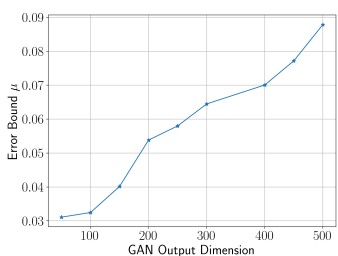

**Random GAN.** We first verify Corollary 4 when $W$ is a random matrix. In accordance with our theory, we run our alternating optimization algorithm for 10 test examples and observe that the iterates always converge to the global optimizer.

Moreover, Figure 1 shows how expansiveness of the GAN affects the local error bound parameter $\mu$. Many existing results on random GAN inversion assume expansiveness of the GAN ($m \gg d$) to prove a benign optimization landscape. By examining $\mu$ instead, our results offer a different viewpoint. Recall that our convergence theory (Theorem 6) shows that as $\mu$ increases, we expect faster convergence. Thus, Figure 1 gives further evidence that expansiveness helps optimization.

Figure 1: We show the output dimension $m$ vs the computed $\mu$ averaged over the optimization path of 10 test examples for a GAN with random weights and latent space dimension $d = 10$.

**Non-Random GAN.** To illustrate an example of a non-random network that can still satisfy the local error bound condition, consider a GAN with latent space dimension $d = 2$ and output dimension $m = 100$. Suppose that the rows of $W$ are spanned by two orthonormal vectors $\begin{bmatrix} -\sqrt{2}/2 & \sqrt{2}/2 \end{bmatrix}$ and $\begin{bmatrix} \sqrt{2}/2 & \sqrt{2}/2 \end{bmatrix}$. The distribution of these rows is far from the uniform distribution on the unit sphere, and also does not satisfy the Weight Distribution Condition (WDC) from Corollary 4 for small values of $\epsilon$ [4]. The optimization landscape is still benign, and we can reliably converge to the global optimum. Further, it satisfies the local error bound condition empirically (details in Appendix). Since $d = 2$, we plot the landscape for the GAN inversion problem when we set $c_a^* = c_a^k = 0$ - Figure 2 confirms the benign landscape. Examples of more non-random networks and corresponding values of $\mu$ can be found in the Appendix.

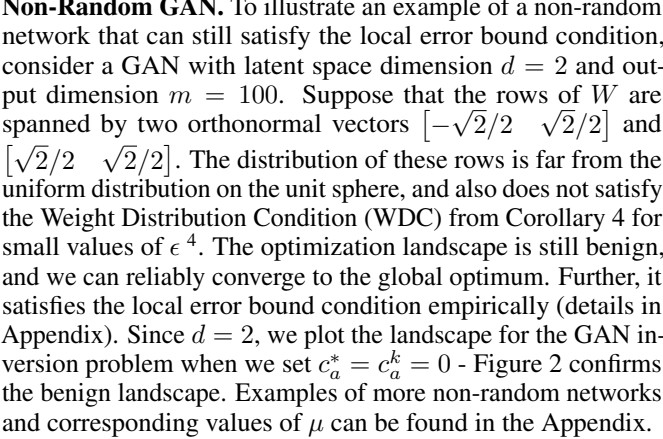

Figure 2: Optimization landscape for 2-D GAN inversion problem with weights spanned by orthonormal vectors. See text for details.

---

[4]in Huang et al. (2021), $\epsilon$ must be less than $\frac{1}{d^{90}}$ which is a very small number even for $d = 2$

## 5.2 Reverse Engineering of Deceptions on Real Data

**Experimental Setup.** We consider the family of $\{\ell_1, \ell_2, \ell_\infty\}$ PGD attacks - the full experimental details of the attacks and network architectures can be found in the Appendix. We use a pretrained DCGAN, Wasserstein-GAN, and StyleGAN-XL for the MNIST, Fashion-MNIST and CIFAR-10 datasets respectively (Radford et al., 2015; Arjovsky et al., 2017; Sauer et al., 2022; LeCun, 1998; Xiao et al., 2017; Krizhevsky et al., 2009). The attack dictionary $D_a$ contains $\ell_p$ attacks for $p \in \{1, 2, \infty\}$ evaluated on 200 training examples per class. It is divided into blocks where each block corresponds to a signal class and attack type pair, i.e., block $(i, j)$ of $D_a$ denotes signal class $i$ and $\ell_p$ attack type $j$.

**Signal Classification Baselines.** We emphasize that the RED problem tries to jointly classify signal and threat model in a principled manner. However, we can still compare the signal classification approach to adversarial training designed to defend against a union of threat models. The first baselines are $M_1, M_2$ and $M_\infty$, which are adversarial training algorithms for $\ell_1, \ell_2$ and $\ell_\infty$ attacks respectively. We then compare to the SOTA specialized adversarial training algorithm MSD (Maini et al., 2020; Tramer & Boneh, 2019). Lastly, we compare to the structured block-sparse classifier (SBSC) from Thaker et al. (2022), which relies on the linear subspaces assumption on the data.

**Attack Classification Baselines.** Our main comparison is to Thaker et al. (2022), which is another principled attack detector, denoted as the structured block-sparse attack detector (SBSAD). While there are other works that study the RED problem, there is no standardized evaluation protocol for the RED problem yet, and other works focus on different problems such as classifying attacks in the low-data regime or recovering exact parameters of PGD attacks.

**Algorithm.** To classify the signal and attack for an adversarial example $x'$ computed on classification network $\psi$, we run Algorithm 1. We initialize $z^*$ to the solution of the Defense-GAN method applied to $x'$, which runs GAN inversion on $x'$ directly (Samangouei et al., 2018). Our methods are:

1. BSD-GAN (Block-Sparse Defense GAN): The signal classifier that runs Algorithm 1 and then uses $G(z^k)$ as input to the classification network $\psi$ to generate a label.

2. BSD-GAN-AD (Block-Sparse Defense GAN Attack Detector): This method returns the block $\hat{j}$ of the attack dictionary $D_a$ that minimizes reconstruction loss $\left\| x' - G(z^k) - D_a[i][\hat{j}]c_a[i][\hat{j}] \right\|_2$ for all $i$.

Further experimental details such as step sizes and initialization details can be found in the Appendix. We emphasize that the key strength of our model is that any generative model can be used to model the clean data as long as it has favorable inversion properties. Given the wide successes of inverting generative models in various applications, we believe our approach generalizes to the rich set of applications where GAN inversion is used. Our goal in this section is to show that our algorithm improves upon existing approaches for the RED problem while still providing theoretical guarantees.

**Remark about assumptions.** Recall that our theoretical analysis requires two assumptions, Assumption 1 on the smoothness of the activation function of the GAN, and Assumption 5, a local error bound condition. In the previous section, we showed that Assumption 5 holds for many networks. In this section, we relax Assumption 1 to consider a variety of practical pretrained GANs. In Appendix, we show the approach empirically works in the setting of our theoretical results as well.

Table 1: Adversarial image and attack classification accuracy on digit classification of MNIST dataset. Refer to main text for acronym descriptions.

| **MNIST** | CNN | $M_\infty$ | $M_2$ | $M_1$ | MSD | SBSC | BSD-GAN | SBSAD | BSD-GAN-AD |
|---|---|---|---|---|---|---|---|---|---|
| Clean accuracy | 98.99% | 99.1% | 99.2% | 99.0% | 98.3% | 92% | 94% | - | - |
| $\ell_\infty$ PGD ($\epsilon = 0.3$) | 0.03% | **90.3%** | 0.4% | 0.0% | 62.7% | 77.27% | 75.3% | 73.2% | **92.3%** |
| $\ell_2$ PGD ($\epsilon = 2.0$) | 44.13% | 68.8% | 69.2% | 38.7% | 70.2% | 85.34% | **89.6%** | 46% | **63%** |
| $\ell_1$ PGD ($\epsilon = 10.0$) | 41.98% | 61.8% | 51.1% | 74.6% | 70.4% | 85.97% | **87.8%** | 36.6% | **95.8%** |
| Average | 28.71% | 73.63% | 40.23% | 37.77% | 67.76% | 82.82% | **84.23%** | 51.93% | **83.7%** |

### 5.2.1 MNIST AND FASHION-MNIST

For the MNIST and Fashion-MNIST datasets, we expect that the method from Thaker et al. (2022) will not work since the data does not lie in a union of linear subspaces. For the MNIST dataset, Table 1 illustrates first that surprisingly, even the baselines from Thaker et al. (2022) are better than the adversarial training baselines at signal classification. However, our approach improves upon this method since the GAN is a better model of the clean data distribution. The improved data model results in not only higher signal classification accuracy on average, but also significantly higher attack classification accuracy since the signal error is lower. We also observe that discerning between $\ell_2$ and $\ell_1$ attacks is difficult, a phenomenon consistent with other works on the RED problem (Moayeri & Feizi, 2021; Thaker et al., 2022). We defer the experiments on the Fashion-MNIST dataset to the Appendix, where we observe similar conclusions to performance on the MNIST dataset.

### 5.2.2 CIFAR-10

We use a class-conditional StyleGAN-XL to model the clean CIFAR-10 data and a WideResnet as the classification network, which achieves 96% clean test accuracy. Inspired by other StyleGAN inversion works (Abdal et al., 2019), we invert in the $\mathcal{W}+$ space (the space generated after the mapping network). We initialize the iterates of the GAN inversion problem to a vector in $\mathcal{W}+$ that is generated by the mapping network applied to a random $z$ and a random class. Interestingly, the GAN inversion problem usually converges to an image of the correct class regardless of the class of the initialization, suggesting a benign landscape of the class-conditional StyleGAN.

Our results in Table 6 show a $\approx 60\%$ improvement in signal classification accuracy on CIFAR-10 using the GAN model as opposed to the model from Thaker et al. (2022). The attack classification accuracy also improves on average from 37% to 56% compared to the model that uses the linear subspace assumption for the data. However, for $\ell_\infty$ and $\ell_1$ attacks, we do not observe very high attack classification accuracy. We conjecture that this is due to the complexity of the underlying WideResnet (Zagoruyko & Komodakis, 2016). Namely, the results of Thaker et al. (2022) show that the attack dictionary model is valid only for fully connected locally linear networks. Extending the attack model to handle a wider class of networks is an important future direction.

Table 2: Adversarial image and attack classification accuracy on CIFAR-10 dataset for 100 test examples. See Table 1 for column descriptions.

| **CIFAR-10** | CNN | SBSC | BSD-GAN | SBSAD | BSD-GAN-AD |
|---|---|---|---|---|---|
| Clean accuracy | 99% | 52% | 72% | - | - |
| $\ell_\infty$ PGD ($\epsilon = 0.03$) | 0% | 15% | **76%** | 14% | **48%** |
| $\ell_2$ PGD ($\epsilon = 0.5$) | 0% | 18% | **87%** | 36% | **77%** |
| $\ell_1$ PGD ($\epsilon = 12.0$) | 0% | 18% | **71%** | **63%** | 44% |
| Average | 0% | 17% | **78%** | 37.66% | **56%** |

## 6 CONCLUSION

In this paper, we proposed a GAN inversion-based approach to reverse engineering adversarial attacks with provable guarantees. In particular, we relax assumptions in prior work that clean data lies in a union of linear subspaces to instead leverage the power of nonlinear deep generative models to model the data distribution. For the corresponding nonconvex inverse problem, under local error bound conditions, we demonstrated linear convergence to global optima. Finally, we empirically demonstrated the strength of our model on the MNIST, Fashion-MNIST, and CIFAR-10 datasets. We believe our work has many promising future directions such as verifying the local error bound conditions theoretically as well as relaxing them further to understand the benign optimization landscape of inverting deep generative models.

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

APPENDIX

# A  PROOFS FOR THEORETICAL RESULTS

For pedagogical purposes and to highlight our main results for the challenging nonconvex GAN inversion problem, the main text is structured in order of increasing complexity of the problem (first with GAN inversion, then with unregularized RED problem, and finally with regularized RED problem). However, the proofs for GAN inversion and unregularized RED problem are better understood when viewing GAN inversion as a special case of the unregularized RED problem with $c_a = 0$. As such, we begin by proving the more general result of the unregularized RED problem from Section 4.2. We will then take the main theorem from Section 4.1 as a special case of this general result.

## A.1  PROOFS FOR SECTION 4.2: REVERSE ENGINEERING OF DECEPTIONS OPTIMIZATION PROBLEM WITHOUT REGULARIZATION

Recall that we formulate an inverse problem to learn $z$ and $c_a$:

$$\min_{z,c_a} \mathcal{L}(z, c_a) \triangleq f(z, c_a) + \lambda h(c_a), \tag{13}$$

where $f(z, c_a) = \|x' - G(z) - D_a c_a\|_2^2$ denotes a reconstruction loss and $h(c_a)$ denotes a (nonsmooth) convex regularizer on the coefficients $c_a$.

The proof strategy for our main theorem in Section 4.2 relies mainly on an almost co-coercivity of the gradient (Lemma 9), which we show next.

**Lemma 9.** *(Almost co-coercivity) We have that the gradient operator of $f(z, c_a)$ is almost co-coercive i.e.*

$$\langle \nabla_{c_a} f(z^k, c_a^k) - \nabla_{c_a} f(z^*, c_a^*), c_a^k - c_a^* \rangle + \langle \nabla_z f(z^k, c_a^k) - \nabla_z f(z^*, c_a^*), z^k - z^* \rangle \geq \tag{14}$$

$$2\eta \left(1 - \frac{\eta\rho}{2}\right) \left[ \left\| \nabla_z f(z^k, c_a^k) - \nabla_z f(z^*, c_a^*) \right\|_2^2 + \left\| \nabla_{c_a} f(z^k, c_a^k) - \nabla_{c_a} f(z^*, c_a^*) \right\|_2^2 \right]$$

$$- \epsilon [ \left\| z^k - z^* - \eta(\nabla_z f(z^k, c_a^k) - \nabla_z f(z^*, c_a^*)) \right\|_2^2$$

$$+ \left\| c_a^k - c_a^* - \eta(\nabla_{c_a} f(z^k, c_a^k) - \nabla_{c_a} f(z^*, c_a^*)) \right\|_2^2 ]$$

*where $\rho$ and $-\epsilon$ denote the maximum and minimum eigenvalues of the Hessian of the loss respectively.*

*Proof.* We note that the proof of this result is adapted from Richards & Kuzborskij (2021), Lemma 5. The key differences are that we do not consider a stability of iterates when changing one datapoint as in Richards & Kuzborskij (2021), but rather show an almost co-coercivity of the gradient across iterates of our gradient descent algorithm. Further, our analysis requires extra assumptions such as the local error bound condition in order to demonstrate convergence of the iterates beyond this lemma.

We wish to lower bound $\langle \nabla_{c_a} f(z^k, c_a^k) - \nabla_{c_a} f(z^*, c_a^*), c_a^k - c_a^* \rangle + \langle \nabla_z f(z^k, c_a^k) - \nabla_z f(z^*, c_a^*), z^k - z^* \rangle$. We can rewrite this inner product in a different way using the functions:

$$\psi(z, c_a) \triangleq f(z, c_a) - \langle \nabla_z f(z^*, c_a^*), z \rangle - \langle \nabla_{c_a} f(z^*, c_a^*), c_a \rangle \tag{15}$$

$$\psi^\star(z, c_a) \triangleq f(z, c_a) - \langle \nabla_z f(z^k, c_a^k), z \rangle - \langle \nabla_{c_a} f(z^k, c_a^k), c_a \rangle \tag{16}$$

Then, some simple algebra shows that:

$$\langle \nabla_{c_a} f(z^k, c_a^k) - \nabla_{c_a} f(z^*, c_a^*), c_a^k - c_a^* \rangle + \langle \nabla_z f(z^k, c_a^k) - \nabla_z f(z^*, c_a^*), z^k - z^* \rangle \tag{17}$$

$$= \psi(z^k, c_a^k) - \psi(z^*, c_a^*) + \psi^\star(z^*, c_a^*) - \psi^\star(z^k, c_a^k) \tag{18}$$

Now, we will bound $\psi(z^k, c_a^k) - \psi(z^*, c_a^*)$ and $\psi^\star(z^*, c_a^*) - \psi^\star(z^k, c_a^k)$ separately.

We prove it for $\psi(z^k, c_a^k) - \psi(z^*, c_a^*)$ and the proofs are symmetric replacing $\psi$ with $\psi^\star$. The proof strategy will be to upper and lower bound a different term, namely $\psi(z^k - \eta\nabla_z\psi(z^k, c_a^k), c_a^k - \eta\nabla_{c_a}\psi(z^k, c_a^k))$.

We begin with the upper bound, which uses $\rho$-smoothness of the loss and Taylor's approximation to give:

$$\psi(z^k - \eta\nabla_z\psi(z^k, c_a^k), c_a^k - \eta\nabla_{c_a}\psi(z^k, c_a^k)) \leq \psi(z^k, c_a^k) \tag{19}$$

$$- \eta\left(1 - \frac{\eta\rho}{2}\right)\left(\left\|\nabla_z\psi(z^k, c_a^k)\right\|_2^2 + \left\|\nabla_{c_a}\psi(z^k, c_a^k)\right\|_2^2\right) \tag{20}$$

The lower bound is a bit more tricky (we normally would just use convexity and smoothness to lower bound by $\psi(z^k, c_a^k)$). We start by defining the following quantities:

$$z(\alpha) = z^* + \alpha(z^k - z^* - \eta(\nabla_z f(z^k, c_a^k) - \nabla_z f(z^*, c_a^*))) \tag{21}$$

$$c_a(\alpha) = c_a^* + \alpha(c_a^k - c_a^* - \eta(\nabla_{c_a} f(z^k, c_a^k) - \nabla_{c_a} f(z^*, c_a^*)) \tag{22}$$

We then define a function $g(\alpha)$ as:

$$g(\alpha) = \psi(z(\alpha), c_a(\alpha)) + \frac{\epsilon\alpha^2}{2}(\|\zeta_z\|_2^2 + \|\zeta_{c_a}\|_2^2) \tag{23}$$

where $\zeta_z = z^k - z^* - \eta(\nabla_z f(z^k, c_a^k) - \nabla_z f(z^*, c_a^*))$ and $\zeta_{c_a} = c_a^k - c_a^* - \eta(\nabla_{c_a} f(z^k, c_a^k) - \nabla_{c_a} f(z^*, c_a^*))$ Now, we have that $g''(\alpha) \geq 0$ since $-\epsilon$ is the smallest eigenvalue of the Hessian of the loss, and using the following expansion of $g''(\alpha)$:

$$g'(\alpha) = \begin{bmatrix}\zeta_z & \zeta_{c_a}\end{bmatrix}^T (\nabla f(z(\alpha), c_a(\alpha)) - \nabla_z f(z^*, c_a^*) - \nabla_{c_a} f(z^*, c_a^*)) + \epsilon\alpha(\|\zeta_z\|_2^2 + \|\zeta_{c_a}\|_2^2) \tag{24}$$

$$g''(\alpha) = \begin{bmatrix}\zeta_z & \zeta_{c_a}\end{bmatrix}^T \nabla^2 f(z(\alpha), c_a(\alpha)) \begin{bmatrix}\zeta_z & \zeta_{c_a}\end{bmatrix} + \epsilon(\|\zeta_z\|_2^2 + \|\zeta_{c_a}\|_2^2) \tag{25}$$

This shows that $g$ is convex, so we have that $g(1) - g(0) \geq g'(0) = 0$. Plugging in the definition of $g(1)$ and $g(0)$, we have:

$$\psi(z^k - \eta\nabla_z\psi(z^k, c_a^k), c_a^k - \eta\nabla_{c_a}\psi(z^k, c_a^k)) \geq \psi(z^*, c_a^*) - \frac{\epsilon}{2}\left(\|\zeta_z\|_2^2 + \|\zeta_{c_a}\|_2^2\right) \tag{26}$$

Rearranging the above lower and upper bounds, we are left with a lower bound on $\psi(z^k, c_a^k) - \psi(z^*, c_a^*)$ as desired. As mentioned previously, the same arguments hold above to give a lower bound on $\psi^\star(z^*, c_a^*) - \psi^\star(z^k, c_a^k)$. This gives us our final bound, which is that:

$$\langle\nabla_{c_a} f(z^k, c_a^k) - \nabla_{c_a} f(z^*, c_a^*), c_a^k - c_a^*\rangle + \langle\nabla_z f(z^k, c_a^k) - \nabla_z f(z^*, c_a^*), z^k - z^*\rangle \geq \tag{27}$$

$$2\eta\left(1 - \frac{\eta\rho}{2}\right)\left[\left\|\nabla_z f(z^k, c_a^k) - \nabla_z f(z^*, c_a^*)\right\|_2^2 + \left\|\nabla_{c_a} f(z^k, c_a^k) - \nabla_{c_a} f(z^*, c_a^*)\right\|_2^2\right]$$

$$- \epsilon[\left\|z^k - z^* - \eta(\nabla_z f(z^k, c_a^k) - \nabla_z f(z^*, c_a^*))\right\|_2^2$$

$$+ \left\|c_a^k - c_a^* - \eta(\nabla_{c_a} f(z^k, c_a^k) - \nabla_{c_a} f(z^*, c_a^*))\right\|_2^2]$$

$$\square$$

Another important lemma before proving our main result is the following.

**Lemma 10.** *Define the following quantities:*

$$\zeta_z \triangleq z^k - z^* - \eta(\nabla_z f(z^k, c_a^k) - \nabla_z f(z^*, c_a^*)) \tag{28}$$

$$\zeta_{c_a} \triangleq c_a^k - c_a^* - \eta(\nabla_{c_a} f(z^k, c_a^k) - \nabla_{c_a} f(z^*, c_a^*)) \tag{29}$$

*Then, assuming that $\eta < \frac{1}{2\epsilon}$ and $\eta < \frac{3}{2\rho}$ , we have:*

$$\|\zeta_z\|_2^2 + \|\zeta_{c_a}\|_2^2 \leq \frac{1}{1 - 2\eta\epsilon}(\|\Delta z^k\|_2^2 + \|\Delta c_a^k\|_2^2) \tag{30}$$

*Proof.* We will expand the definition of $\zeta_z$ and $\zeta_{c_a}$ and use co-coercivity (Lemma 9) again. This gives us that $\|\zeta_z\|_2^2 + \|\zeta_{c_a}\|_2^2$ is equal to:

$$= \|\Delta z^k\|_2^2 + \|\Delta c_a^k\|_2^2 \tag{31}$$

$$+ \eta^2 \left[ \|\nabla_z f(z^k, c_a^k) - \nabla_z f(z^*, c_a^*)\|_2^2 + \|\nabla_{c_a} f(z^k, c_a^k) - \nabla_{c_a} f(z^*, c_a^*)\|_2^2 \right] \tag{32}$$

$$- 2\eta \left[ \langle \nabla_{c_a} f(z^k, c_a^k) - \nabla_{c_a} f(z^*, c_a^*), c_a^k - c_a^* \rangle + \langle \nabla_z f(z^k, c_a^k) - \nabla_z f(z^*, c_a^*), z^k - z^* \rangle \right] \tag{33}$$

$$\leq \|\Delta z^k\|_2^2 + \|\Delta c_a^k\|_2^2 \tag{34}$$

$$+ \eta^2 \left[ \|\nabla_z f(z^k, c_a^k) - \nabla_z f(z^*, c_a^*)\|_2^2 + \|\nabla_{c_a} f(z^k, c_a^k) - \nabla_{c_a} f(z^*, c_a^*)\|_2^2 \right] \tag{35}$$

$$- 4\eta^2 \left(1 - \frac{\eta\rho}{2}\right) \left[ \|\nabla_z f(z^k, c_a^k) - \nabla_z f(z^*, c_a^*)\|_2^2 + \|\nabla_{c_a} f(z^k, c_a^k) - \nabla_{c_a} f(z^*, c_a^*)\|_2^2 \right] \tag{36}$$

$$+ 2\eta\epsilon \left[ \|\zeta_z\|_2^2 + \|\zeta_{c_a}\|_2^2 \right] \tag{37}$$

$$\leq \left(1 + \eta^2\rho^2 - 4\eta^2\rho^2 \left(1 - \frac{\eta\rho}{2}\right)\right) \left[ \|\Delta z^k\|_2^2 + \|\Delta c_a^k\|_2^2 \right] + 2\eta\epsilon \left[ \|\zeta_z\|_2^2 + \|\zeta_{c_a}\|_2^2 \right] \tag{38}$$

If $\eta < \frac{1}{2\epsilon}$, we can rearrange this inequality such that

$$\|\zeta_z\|_2^2 + \|\zeta_{c_a}\|_2^2 \leq \frac{1}{1 - 2\eta\epsilon} \left(1 + \eta^2\rho^2 - 4\eta^2\rho^2 \left(1 - \frac{\eta\rho}{2}\right)\right) \left[ \|\Delta z^k\|_2^2 + \|\Delta c_a^k\|_2^2 \right] \tag{39}$$

Lastly, using the assumption that $\eta < \frac{3}{2\rho}$, we have that $\eta^2\rho^2 - 4\eta^2\rho^2 \left(1 - \frac{\eta\rho}{2}\right) < 0$, so we can drop it from the equation above and have a simpler upper bound:

$$\|\zeta_z\|_2^2 + \|\zeta_{c_a}\|_2^2 \leq \frac{1}{1 - 2\eta\epsilon}(\|\Delta z^k\|_2^2 + \|\Delta c_a^k\|_2^2) \tag{40}$$

$\square$

Now, we can prove our main result, Theorem 6 restated below.

**Theorem 11.** *Suppose that Assumption 1 holds for the nonlinear activation function and Assumption 5 holds with local error bound parameter $\mu$. Let $\rho$ and $-\epsilon$ be the maximum and minimum eigenvalues of the Hessian of the loss. Further, assume that the step size satisfies $\eta \leq \min\left\{\frac{1}{4\epsilon}, \frac{3}{2\rho}\right\}$ and $\eta \in \left(\frac{3\mu^2 - \sqrt{9\mu^4 - 32\mu^2\rho\epsilon}}{4\mu^2\rho}, \frac{3\mu^2 + \sqrt{9\mu^4 - 32\mu^2\rho\epsilon}}{4\mu^2\rho}\right)$. Lastly, assume that $\mu \gtrsim \sqrt{\rho\epsilon}$. Then, we have that the iterates converge linearly to the global optimum with the following rate in $(0, 1)$:*

$$\|\Delta z^{k+1}\|_2^2 + \|\Delta c_a^{k+1}\|_2^2 \leq \left(1 - 4\eta^2\mu^2 \left(\frac{3}{4} - \frac{\eta\rho}{2}\right) + 4\eta\epsilon\right) (\|\Delta z^k\|_2^2 + \|\Delta c_a^k\|_2^2) \tag{41}$$

*Proof.* We can expand the suboptimality of the iterates $\left\|\Delta z^{k+1}\right\|_2^2 + \left\|\Delta c_a^{k+1}\right\|_2^2$ at the $k+1$ iteration of gradient descent as follows:

$$= \left\|c_a^k - \eta\nabla_{c_a}f(z^k, c_a^k) - c_a^* - \eta\nabla_{c_a}f(z^*, c_a^*)\right\|_2^2 \tag{42}$$

$$+ \left\|z^k - \eta\nabla_z f(z^k, c_a^k) - z^* + \eta\nabla_z f(z^*, c_a^*)\right\|_2^2 \tag{43}$$

$$\leq \left\|c_a^k - c_a^*\right\|_2^2 - 2\eta\langle\nabla_{c_a}f(z^k, c_a^k) - \nabla_{c_a}f(z^*, c_a^*), c_a^k - c_a^*\rangle \tag{44}$$

$$+ \eta^2\left\|\nabla_{c_a}f(z^k, c_a^k) - \nabla_{c_a}f(z^*, c_a^*)\right\|_2^2 \tag{45}$$

$$+ \left\|z^k - z^*\right\|_2^2 - 2\eta\langle\nabla_z f(z^k, c_a^k) - \nabla_z f(z^*, c_a^*), z^k - z^*\rangle \tag{46}$$

$$+ \eta^2\left\|\nabla_z f(z^k, c_a^k) - \nabla_z f(z^*, c_a^*)\right\|_2^2 \tag{47}$$

$$\overset{\text{(Lemma 9)}}{\leq} \left\|\Delta z^k\right\|_2^2 + \left\|\Delta c_a^k\right\|_2^2 \tag{48}$$

$$- 4\eta^2\left(\frac{3}{4} - \frac{\eta\rho}{2}\right)\left\|\nabla_z f(z^k, c_a^k) - \nabla_z f(z^*, c_a^*)\right\|_2^2 \tag{49}$$

$$- 4\eta^2\left(\frac{3}{4} - \frac{\eta\rho}{2}\right)\left\|\nabla_{c_a}f(z^k, c_a^k) - \nabla_{c_a}f(z^*, c_a^*)\right\|_2^2 \tag{50}$$

$$+ 2\eta\epsilon\left\|z^k - z^* - \eta(\nabla_z f(z^k, c_a^k) - \nabla_z f(z^*, c_a^*))\right\|_2^2 \tag{51}$$

$$+ 2\eta\epsilon\left\|c_a^k - c_a^* - \eta(\nabla_{c_a}f(z^k, c_a^k) - \nabla_{c_a}f(z^*, c_a^*))\right\|_2^2 \tag{52}$$

$$\overset{\text{(Error Bound)}}{\leq} \left\|\Delta z^k\right\|_2^2 + \left\|\Delta c_a^k\right\|_2^2 \tag{53}$$

$$- 4\eta^2\mu^2\left(\frac{3}{4} - \frac{\eta\rho}{2}\right)(\left\|\Delta z^k\right\|_2^2 + \left\|\Delta c_a^k\right\|_2^2) \tag{54}$$

$$+ 2\eta\epsilon\left\|z^k - z^* - \eta(\nabla_z f(z^k, c_a^k) - \nabla_z f(z^*, c_a^*))\right\|_2^2 \tag{55}$$

$$+ 2\eta\epsilon\left\|c_a^k - c_a^* - \eta(\nabla_{c_a}f(z^k, c_a^k) - \nabla_{c_a}f(z^*, c_a^*))\right\|_2^2 \tag{56}$$

$$\overset{\text{(Lemma 10)}}{\leq} \left(1 - 4\eta^2\mu^2\left(\frac{3}{4} - \frac{\eta\rho}{2}\right) + \frac{2\eta\epsilon}{1 - 2\eta\epsilon}\right)(\left\|\Delta z^k\right\|_2^2 + \left\|\Delta c_a^k\right\|_2^2) \tag{57}$$

When $\eta < \frac{1}{4\epsilon}$, the last term in the rate $\frac{2\eta\epsilon}{1-2\eta\epsilon}$ is upper bounded by $4\eta\epsilon$. We now examine when this rate is less than 1. This is equivalent to showing that

$$-4\eta^2\mu^2\left(\frac{3}{4} - \frac{\eta\rho}{2}\right) + 4\eta\epsilon < 0 \tag{58}$$

Factoring out a factor of $\eta$, we are left with a quadratic in $\eta$. We need the discriminant of this quadratic to be positive in order to have real roots. This gives us a condition that (dropping constant factors):

$$\mu \gtrsim \sqrt{\rho\epsilon} \tag{59}$$

The roots of this quadratic in $\eta$ give us a range where the rate is less than 1. Thus, we require the the step size to be in this range for convergence:

$$\eta \in \left(\frac{3\mu^2 - \sqrt{9\mu^4 - 32\mu^2\rho\epsilon}}{4\mu^2\rho}, \frac{3\mu^2 + \sqrt{9\mu^4 - 32\mu^2\rho\epsilon}}{4\mu^2\rho}\right) \tag{60}$$

$\square$

## A.2 PROOFS FOR SECTION 4.1: GAN INVERSION

The proof of the main result of this section, Theorem 3, is identical to the proof of the main theorem above by taking $c_a^k = c_a^* = 0$.

Next, we elaborate on the comparison in Section 4.1.1 to existing works on the GAN inversion problem. First, the following definition restates the WDC condition from Hand & Voroninski (2017) for completeness.

**Definition 12.** *(Weight Distribution Condition Hand & Voroninski (2017)) A matrix $W \in \mathbb{R}^{n \times k}$ satisfies the Weight Distribution Condition (WDC) with constant $\epsilon$ if for all nonzero $x, y \in R^k$,*

$$\left\| \sum_{i=1}^n \sigma'(w_i \cdot x)\sigma'(w_i \cdot y) \cdot w_i w_i^t - Q_{x,y} \right\|_2 \leq \epsilon \tag{61}$$

*with $Q_{x,y} = \mathbb{E}[\sigma'(w_i \cdot x)\sigma'(w_i \cdot y) \cdot w_i w_i^t]$ for $w_i \sim N(0, I_k/n)$.*

### A.2.1 PROOF OF COROLLARY 4

*Proof.* From Theorem 2 in Hand & Voroninski (2017), we have that when the conditions of the corollary are met, then there exists a direction $v$ such that the directional derivative of $f(z^0)$ in direction $v$ is less than 0 when

$$z^0 \notin \mathcal{B}(z^*, K_2 L^3 \epsilon^{1/4} \|z^*\|_2) \cup \mathcal{B}(-\kappa z^*, K_2 L^{13} \epsilon^{1/4} \|z^*\|_2) \cup \{0\} \tag{62}$$

This implies that $z^0$ is not a stationary point and further that there exists a descent direction at that point. Thus, there must exist a $\mu > 0$ such that the local error bound condition holds at $z^0$. For example, when $L = 1$ and when the activation function has first derivative bounded away from 0, then this value of $\mu$ will simply be the minimum singular value of $W_1^T R_{z^0}$ where $R_{z_0} = \mathrm{diag}(\sigma'(W_1 z^0))$. The authors of Huang et al. (2021) demonstrate that for a subgradient descent algorithm, the iterates stay out of the basin of attraction for the spurious stationary point, and a descent direction still exists. Thus, along the optimization trajectory, the local error bound condition holds following the same logic as above. □

## A.3 PROOF OF THEOREM 8: REGULARIZED CASE

*Proof.* First, we note that because the function $f$ is $\rho$-smooth by assumption, we have that for all $(z, c_a)$ and $(\tilde{z}, \tilde{c}_a)$:

$$f(z, c_a) \leq f(\tilde{z}, \tilde{c}_a) + \langle \nabla f(\tilde{z}, \tilde{c}_a), [z - \tilde{z} \quad c_a - \tilde{c}_a] \rangle + \frac{\rho}{2} \left\| [z - \tilde{z} \quad c_a - \tilde{c}_a] \right\|_2^2 \tag{63}$$

Next, we expand the loss:

$$\mathcal{L}(z^{k+1}, c_a^{k+1}) = f(z^{k+1}, c_a^{k+1}) + h(c_a^{k+1}) + h(c_a^k) - h(c_a^k) \tag{64}$$

$$= f(z^k, c_a^k) + h(c_a^k) + \langle \nabla f(z^k, c_a^k), \begin{bmatrix} z^{k+1} - z^k & c_a^{k+1} - c_a^k \end{bmatrix} \rangle \tag{65}$$

$$+ \frac{\rho}{2} \left\| \begin{bmatrix} z^{k+1} - z^k & c_a^{k+1} - c_a^k \end{bmatrix} \right\|_2^2 + h(c_a^{k+1}) - h(c_a^k)$$

$$= \mathcal{L}(z^k, c_a^k) + \min_y \left[ \langle \nabla_{c_a} f(z^k, c_a^k), y - c_a^k \rangle + \frac{\rho}{2} \left\| y - c_a^k \right\|_2^2 + h(y) - h(c_a^k) \right] \tag{66}$$

$$+ \langle \nabla_z f(z^k, c_a^k), z^{k+1} - z^k \rangle + \frac{\rho}{2} \left\| z^{k+1} - z^k \right\|_2^2$$

$$\leq \mathcal{L}(z^k, c_a^k) + \min_y \left[ \langle \nabla_{c_a} f(z^k, c_a^k), y - c_a^k \rangle + \frac{\rho}{2} \left\| y - c_a^k \right\|_2^2 + h(y) - h(c_a^k) \right] \tag{67}$$

$$- \eta \left\| \nabla_z f(z^k, c_a^k) \right\|_2^2 + \frac{\eta^2 \rho}{2} \left\| \nabla_z f(z^k, c_a^k) \right\|_2^2$$

$$\leq \mathcal{L}(z^k, c_a^k) - \frac{\mu}{\rho} (\mathcal{L}(z^k, c_a^k) - \mathcal{L}(z^*, c_a^*)) \tag{68}$$

This implies our final result:

$$\mathcal{L}(z^{k+1}, c_a^{k+1}) - \mathcal{L}(z^*, c_a^*) \leq \left( 1 - \frac{\mu}{\rho} \right) (\mathcal{L}(z^k, c_a^k) - \mathcal{L}(z^*, c_a^*)) \tag{69}$$

$$\square$$

### A.4 INSTANTIATING $\rho$ AND $\epsilon$ FOR A SIMPLE NETWORK

When we assume that the network weights and $D_a$ have bounded spectrum as well as the inputs being bounded, we can derive a bound on $\rho$ and $\epsilon$ for a network. For simplicity, we consider a 1-layer network although these arguments will generalize to the $L$-layer case as well. Formally, let $G(z) = \sigma(Wz)$. We use the following assumptions to bound $\rho$ and $\epsilon$:

**Assumption 13.** *(Loss Function and Weights) Assume that $\|x' - G(z) - D_a c_a\|_1 \leq C_0$ for all $z^k, c_a^k$ for an absolute constant $C_0$ i.e. the $\ell_1$ loss is bounded uniformly. This is equivalent to assuming that the inputs $z$ and $c_a$ are bounded. Assume that $\|W_i\|_2 \leq C_{W_i}$ for all $i$, $\sum_{j=1}^{n_i} \|W_i[j]\|_2^2 \leq V_{W_i}$[5], $\|D_a\|_2 \leq C_D$ and $\sigma_{\min}(D_a) > 0$.*

**Lemma 14.** *Fix $z$ and $c_a$. Suppose that Assumptions 1 and 13 hold. Then,*

$$\lambda_{\max}(\nabla^2 f(z, c_a)) \leq \rho \tag{70}$$

$$\min_{\alpha \in [0,1]} \lambda_{\min}(\nabla^2 f(z + \alpha(z^* - z), c_a + \alpha(c_a^* - c_a))) \geq -\epsilon \tag{71}$$

*with $\rho = C_W^2 (B_{\sigma'}^2 + B_{\sigma''} \sqrt{C_0}) + C_W B_{\sigma'} C_D + C_D^2$ and $\epsilon = V_W B_{\sigma''} C_0 + C_W B_{\sigma'} C_D - L_D^2$.*

*Proof.* We begin with noting the Hessian has a block structure due to the two variables:

$$\nabla^2 f(z, c_a) = \begin{bmatrix} \nabla_{z,z} f(z, c_a) & \nabla_{z,c_a} f(z, c_a) \\ \nabla_{c_a, z} f(z, c_a) & \nabla_{c_a, c_a} f(z, c_a) \end{bmatrix} \tag{72}$$

These blocks are equal to:

---

[5]$W_i[j]$ denotes the $j$th row of $W_i$.

$$\nabla_{z,z}[f(z,c_a)] = \nabla_z G(z)\nabla_z G(z)^T + \nabla_{z,z}G(z)(x' - G(z) - D_a c_a) \tag{73}$$

$$\nabla_{z,c_a} f(z,c_a) = \nabla_z G(z)D_a = W^T \sigma'(Wz)D_a \tag{74}$$

$$\nabla_{c_a,z} f(z,c_a) = D_a^T \nabla_z G(z)^T \tag{75}$$

$$\nabla_{c_a,c_a} f(z,c_a) = D_a^T D_a \tag{76}$$

Above, $\nabla_{z,z}G(z)$ is actually a tensor of dimension $d \times d \times m$ since $G$ maps from $\mathbb{R}^d$ to $\mathbb{R}^m$. If we take one slice of this tensor i.e. $\nabla_{z,z}[G(z)]_i$, we will be left with $\sigma''(W_i \cdot z)W_i W_i^T$, where $W_i$ denotes the $i$th row of $W$. We can write this tensor-vector product as the following:

$$\nabla_{z,z}G(z)(x' - G(z) - D_a c_a = \sum_{i=1}^{m} \sigma''(W_i \cdot z)W_i W_i^T \cdot [x' - G(z) - D_a c_a]_i \tag{77}$$

Let $M_{z,z}, M_{z,c_a}, M_{c_a,z}, M_{c_a,c_a}$ be four block matrices corresponding to one nonzero block (conformal to the order in the subscript that gradients are taken) and all the other blocks zero. We can bound the operator norm of the Hessian using triangle inequality:

$$\left\| \nabla^2 f(z,c_a) \right\|_2 = \left\| M_{z,z} + M_{z,c_a} + M_{c_a,z} + M_{c_a,c_a} \right\|_2 \tag{78}$$

$$\leq \left\| M_{z,z} \right\|_2 + \left\| M_{z,c_a} \right\|_2 + \left\| M_{c_a,z} \right\|_2 + \left\| M_{c_a,c_a} \right\|_2 \tag{79}$$

These blocks have operator norm as follows:

$$\left\| M_{z,z} \right\|_2 \leq C_W^2 B_{\sigma'}^2 + 2V_W B_{\sigma''} C_0 \tag{80}$$

$$\left\| M_{z,c_a} \right\|_2 \leq C_W B_{\sigma'} C_D \tag{81}$$

$$\left\| M_{c_a,z} \right\|_2 \leq C_W B_{\sigma'} C_D \tag{82}$$

$$\left\| M_{c_a,c_a} \right\|_2 \leq C_D^2 \tag{83}$$

Plugging into equation 79 yields $\rho$. For the minimum eigenvalue, we have by Weyl's inequality for Hermitian matrices that:

$$\lambda_{\min}(\nabla^2 f(z,c_a)) \geq \lambda_{\min}(M_{z,z}) + \lambda_{\min}\left( \begin{bmatrix} 0 & \nabla_{z,c_a} f(z,c_a) \\ \nabla_{c_a,z} f(z,c_a) & 0 \end{bmatrix} \right) + \lambda_{\min}(M_{c_a,c_a}) \tag{84}$$

$$\geq -\left\| M_{z,z} \right\|_2 - \left( \left\| M_{z,c_a} \right\|_2 + \left\| M_{c_a,z} \right\|_2 \right) + L_D^2 \tag{85}$$

$$\geq -(C_W^2(B_{\sigma'}^2 + 2B_{\sigma''}\sqrt{C_0}) + 2C_W B_{\sigma'} C_D - L_D^2) \tag{86}$$

Note that because the first term in $M_{z,z}$ is PSD, we can remove it from the lower bound, which gives the bound for $\epsilon$ in the theorem. $\square$

## B   ADDITIONAL EXPERIMENTS

### B.1   SYNTHETIC DATA EXPERIMENTAL DETAILS

To set up a realizable problem where the error bound parameter $\mu$ can be computed easily, we use the following setup for a generation of data, a classification network on this data, and a way to compute adversarial attacks given this network.

First, we generate data $x \in \mathbb{R}^m$ from a one-layer GAN $x = G(z)$ with $G(z) = \sigma(Wz)$ and $W \in R^{m \times d}$. We use a leaky RELU activation function as $\sigma$.

Next, we consider a binary classifier on this data of the form $sign(\psi(x))$ where

$$\psi(x) = \frac{1}{\sqrt{k}} \sum_{\ell=1}^{k} a_\ell \sigma(w_\ell \cdot x). \tag{87}$$

Here, the $w_\ell$ are i.i.d from $N\left(0, \frac{1}{m}I_m\right)$ and $a_\ell$ are uniform over $\{-1, 1\}$. We will consider single-step gradient-based attacks $\eta \nabla \psi(x)$. With high probability, a single gradient step will flip the sign of the label, so we can easily find adversarial attacks Bubeck et al. (2021).

To create a realizable instance for the RED problem, we generate a training set $S_{tr} = \{G(z_i) : z_i \sim N(0, I_d)\}_{i=1}^{n_{\text{train}}}$ and similarly a testing set $S_{te}$. The attack dictionary $D_a$ contains single-step gradient attacks on $S_{tr}$. A realizable RED instance is then $x' = x_{te} + D_a c_a^*$ for $x_{te} \in S_{te}$ and some vector $c_a^*$. We run alternating gradient descent as in Section 4.2 to solve for $z^k$ and $c_a^k$. Since we have knowledge of the true $z^*$ and $c_a^*$ for a given problem instance, we can exactly compute the local error bound parameter $\mu$ for a given $z^k$ and $c_a^k$ on the optimization trajectory as $\mu^2 = \frac{\left\|\nabla_z f(z^k, c_a^k)\right\|_2^2 + \left\|\nabla_{c_a} f(z^k, c_a^k)\right\|_2^2}{\left\|\Delta z^k\right\|_2^2 + \left\|\Delta c_a^k\right\|_2^2}$.

## B.2 REAL DATA EXPERIMENTAL DETAILS

| Layer Type | Size |
|---|---|
| Convolution + ReLU | $3 \times 3 \times 32$ |
| Convolution + ReLU | $3 \times 3 \times 32$ |
| Max Pooling | $2 \times 2$ |
| Convolution + ReLU | $3 \times 3 \times 64$ |
| Convolution + ReLU | $3 \times 3 \times 64$ |
| Max Pooling | $2 \times 2$ |
| Fully Connected + ReLU | 200 |
| Fully Connected + ReLU | 200 |
| Fully Connected + ReLU | 10 |

Table 3: Network Architecture for the MNIST and Fashion-MNIST dataset

**Attack Coefficient Algorithm.** In Algorithm 1, we run 500 steps of alternating between updating $z$ and $c_a$. To update $c_a$, we also applied Nesterov acceleration. For the proximal step, we set the step size to be the inverse of the operator norm of $D_a^T D_a$ i.e. the Lipschitz constant of the gradient. To set the regularization parameter, we use the procedure from Thaker et al. (2022) i.e. compute the value of $\lambda$ such that the solution for $c_a$ is the all-zeros vector using the optimality conditions for the problem and then multiplying that value of $\lambda$ by a small constant (e.g. 0.35 for our experiments).

**MNIST.** Table 3 shows the network architecture for the MNIST dataset. This is trained using SGD for 50 epochs with learning rate 0.1, momentum 0.5, and batch size 128, identical to the architecture from Thaker et al. (2022).

All PGD adversaries were generated using the Advertorch library (Ding et al., 2019). We use the same hyperparameters as the adversarial training baselines and as Thaker et al. (2022). Specifically, the $\ell_\infty$ PGD adversary ($\epsilon = 0.3$) used a step size $\alpha = 0.01$ and was run for 100 iterations. The $\ell_2$ PGD adversary ($\epsilon = 2$) used a step size $\alpha = 0.1$ and was run for 200 iterations. The $\ell_1$ PGD adversary ($\epsilon = 10$) used a step size $\alpha = 0.8$ and was run for 100 iterations.

We use a pretrained DCGAN using the architecture from the standard Pytorch implementation (Paszke et al., 2019). The initialization of $z^0$ for Algorithm 1 is as follows: we first sample 10 random initializations for $z$ and for each, run 100 epochs of Defense-GAN training on $x'$ using the MSE loss. Then, we initialize $z^0$ to the vector that gives best MSE loss over the 10 random restarts.

**Fashion-MNIST.** Table 3 shows the network architecture for the Fashion-MNIST dataset. The PGD adversaries have identical hyperparameters as on the MNIST dataset. We use a pretrained Wasserstein-GAN Lindernoren (2023) and use the same initialization scheme as for the MNIST dataset.

**CIFAR-10.** The classification network used is the pretrained Wide Resnet from Pytorch. The $\ell_\infty$ PGD adversary ($\epsilon = 0.03$) used a step size $\alpha = 0.003$ and was run for 100 iterations. The $\ell_2$ PGD adversary ($\epsilon = 0.05$) used a step size $\alpha = 0.05$ and run for 200 iterations. The $\ell_1$ PGD adversary ($\epsilon = 12$) used a step size $\alpha = 1$ and was run for 100 iterations. We used a pretrained StyleGAN-XL (Sauer et al., 2022). To initialize and invert in the space $\mathcal{W}+$, we sampled 10000 initializations of $z$ and one random class. In batch, we run 3000 iterations of Defense-GAN for each initialization and learn a vector in $\mathcal{W}+$ as initialization for the latent space before running Algorithm 1.

**Computational Resources.** All experiments were ran with a single Nvidia GeForce RTX 2080 Ti GPU. For each example, the alternating optimization procedure takes between 1 and 5 minutes depending on the dataset. Note that the optimization procedure for each datapoint can be run in parallel.

### B.3 Local Error Bound for Random Deep Networks

We have run the same experiment as in Figure 1 using a 3-layer GAN to illustrate similar trends as the experiment in the main paper. This GAN is set to be expansive (in accordance with the theoretical results from (Hand & Voroninski, 2017)) e.g. if the input and output dimension are $d$ and $m$ respectively, the weight matrices are $d \times (m/10)$, $(m/5) \times (m/10)$, and $m \times (m/5)$. In Figure 3. We observe a similar trend as in Figure 1 in the paper, in fact the trend is even more exaggerated. We conjecture that the results from Hand & Voroninski (2017) can be re-proven using a language of local error-bound conditions instead, so it is reasonable to expect that it holds for a variety of deep networks as they have shown.

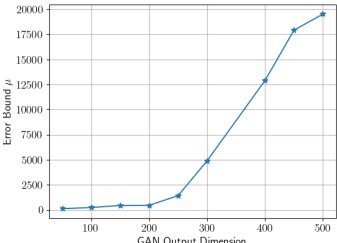

Figure 3: We show the output dimension $m$ vs the computed $\mu$ averaged over the optimization path of 50 test examples for a 3-layer GAN with random weights and input dimension $d = 10$. The layers are expansive with $m/10$ and $m/5$ neurons in the first two layers.

### B.4 On the Restrictiveness of Assumption 1

Our theoretical results assume that the generator network uses a smooth activation functions. In contrast, in our experimental setting, we are given a pre-trained generator network that is not trained with a smooth activation function (the DCGAN, WGAN, and Style-GAN XL all use ReLU or leaky ReLU). In this section, we aim to show that fixing this pretrained network and modifying the activation function to a smooth version of the function does not harm the performance of the generative model or the performance of our signal and attack classifier.

To model this, we use the SoftPlus activation instead of the ReLU activation in our DCGAN and WGAN implementations for the MNIST dataset. The SoftPlus has a temperature parameter $\beta$ which controls how closely the function approximates the ReLU. For our experiments, we use a value of $\beta = 10$. Further, we evaluate these set of experiments on 100 test examples for computational reasons. Our goal is simply to demonstrate that the smoothness assumption on the activation function is a fairly benign assumption needed only for the theoretical results, but does not affect the practicality of the method in practice. Let BSD-GAN denote the approach which uses the nonsmooth ReLU for the generative model. Let BSD-GAN-Smooth denote the approach which uses the Softplus for the generative model (with the pretrained weights of the network fixed to the same values as in BSD-GAN). Similarly, let BSD-GAN-AD and BSD-GAN-AD-Smooth denote the non-smooth and smooth versions of the attack detectors. In Table 4, we demonstrate the gap between the smooth and non-smooth version of the approach, which we call $\Delta_{\text{signal}} := \text{BSD-GAN} - \text{BSD-GAN-Smooth}$

and $\Delta_{\text{attack}} :=$ BSD-GAN-AD $-$ BSD-GAN-AD-Smooth. In all cases, we see a minimal difference between the smooth and non-smooth versions. In fact, on the MNIST dataset, the smooth version often has slightly better accuracy than the non-smooth version for both signal and attack classification. We conjecture one possible reason for this is that for classification, there is strong evidence that networks with smooth activation functions have their own merit (Ramachandran et al., 2017). An interesting direction for future work would be to validate whether generative models can also be similarly trained with smooth activation functions to obtain even better results for the RED problem.

Table 4: Difference between the smooth and non-smooth versions of the method on the MNIST dataset. See text for definition of $\Delta_{\text{signal}}$ and $\Delta_{\text{attack}}$.

| **MNIST** | $\Delta_{\text{signal}}$ | $\Delta_{\text{attack}}$ |
|---|---|---|
| $\ell_\infty$ PGD ($\epsilon = 0.3$) | -1% | 1% |
| $\ell_2$ PGD ($\epsilon = 2.0$) | -1% | -3% |
| $\ell_1$ PGD ($\epsilon = 10.0$) | -1% | -1% |

## B.5  Non-Random Networks that satisfy Local Error Bound Condition

Recall the problem considered in the main text of the paper as a simple example of a non-random network that satisfies the local error bound condition. Consider a GAN with latent space dimension $d = 2$ and output dimension $m = 100$. Suppose that the rows of $W$ are spanned by two orthonormal vectors $\begin{bmatrix} -\sqrt{2}/2 & \sqrt{2}/2 \end{bmatrix}$ and $\begin{bmatrix} \sqrt{2}/2 & \sqrt{2}/2 \end{bmatrix}$. For this problem, with a initialization of $z$ as a standard normal random variable and $c_a$ initialized to the all-zeros vector, we observe an average $\mu$ value of $0.013$ over different random initializations.

We now aim to provide a more general class of non-random networks that satisfy the local error bound condition. We begin with the trivial observation that for a realizable GAN inversion problem, when there is no nonlinearity $\sigma$, any non-random network where $W$ has full (column) rank will satisfy the local error bound condition. This is because the gradient is equal to $\nabla_z f(z^k) = W^T(x' - G(z^k)) = W^T(W\hat{z} - Wz^k)$. Since $W$ is a tall matrix, we have that $W^T W$ is a full-rank matrix and thus the local error bound condition can be satisfied with $\mu$ as the minimum singular value of $W^T W$. However, when we have the nonlinearity, this is not necessarily the case. In this section, we provide several examples of non-random networks that still satisfy the local error bound condition. We provide examples in the 1-layer GAN inversion setting for simplicity i.e. $G(z) = \sigma(Wz)$ with $\sigma$ as the leaky RELU activation function, although these examples although work with the attack dictionary from Section B.1 as well. For all examples below, the ground truth $x$ is generated as $x = G(\hat{z})$ where $\hat{z}$ is drawn from a standard normal.

**Example 15.** *(2-D GAN Inversion) We can slightly modify the example given in Section 5.1 in the following way. Consider a GAN with latent space dimension $d = 2$ and output dimension $m = 100$. Suppose that the rows of $W$ are spanned by $m$ vectors, which are either $\begin{bmatrix} -\sqrt{2}/2 & \sqrt{2}/2 \end{bmatrix} + \epsilon_i$ or $\begin{bmatrix} \sqrt{2}/2 & \sqrt{2}/2 \end{bmatrix} + \epsilon_i$ where $\epsilon_i \sim N(0, 0.2)$ for $i \in [1, \dots, m]$. This is roughly the same distribution as the example in Section 5.1 but with some slight perturbation of $W$ on the 2-d unit sphere. We observe that in this simple case as well, optimization always succeeds to the global minimizer, the landscape looks as benign as in Figure 2, and the average value of $\mu$ computed in practice over 10 test examples is $\mu = 2.17$. Note that the value of $\mu$ is significantly higher than when $\epsilon_i = 0$. We conjecture that the randomness in $\epsilon_i$ leads to an improved landscape as $x' - G(z^k)$ is less likely to fall close to the nullspace of $W_1^T R_1$ (see Algorithm 1 for definition of $R_i$), which is the case when the local error bound condition is not met.*

**Example 16.** *(GAN Inversion with Hadamard Matrices) We can also extend the previous example by looking at Hadamard matrices in general, which are square matrices with entries $1$ and $-1$ and whose rows are mutually orthogonal. Suppose we look at the Hadamard matrix of order $4$, which is:*

$$W = \begin{bmatrix} 1 & 1 & 1 & 1 \\ 1 & -1 & 1 & -1 \\ 1 & 1 & -1 & -1 \\ 1 & -1 & -1 & 1 \end{bmatrix} \tag{88}$$

*Suppose that for a GAN inversion problem with $d = 4, m = 100$, the rows of $W$ are spanned by the 4 rows of this Hadamard matrix. In this case as well, we observe that over 50 runs, GAN inversion always converges to the global minimizer when $z^0$ is randomly initialized. Further, the average value of $\mu$ we observe is $1.07$. This value improves to $\mu = 2.61$ when we add $\epsilon_i$ to each row of $W$ for $\epsilon_i \sim N(0, 0.2)$ as in the previous example. The orthogonality property is likely an important property in ensuring a benign optimization landscape and having the local error bound property hold - note that by construction, the rows are not all mutually orthogonal, but they are spanned by a mutually orthogonal set.*

**Example 17.** *(GAN Inversion with Vandermonde Matrices) Consider a GAN with latent space dimension $d = 2$ and $m = 100$. Let $W$ be a normalized Vandermonde matrix of dimension $m \times d$, i.e. each unnormalized row is $[1 \quad i]$ for $i \in 1, \ldots, m$. For this matrix, over 50 runs, we always converge to the global minimizer with an average $\mu$ value of $0.64$.*

### B.6 FASHION-MNIST EXPERIMENTS

Table 5 gives results of our method on the Fashion-MNIST dataset. we significantly improve upon the work of Thaker et al. (2022) by using a better generative model for the clean data. In particular, it is evident that the clean signal model of Thaker et al. (2022) is insufficient at modelling the clean data, resulting in a clean signal classification accuracy of only 23% on average across different attacks, as compared to our model, which provides 66% test signal classification accuracy.

Table 5: Adversarial image and attack classification accuracy on Fashion-MNIST dataset. See Table 1 for column descriptions.

| Fashion-MNIST | CNN | SBSC | BSD-GAN | SBSAD | BSD-GAN-AD |
|---|---|---|---|---|---|
| $\ell_\infty$ PGD ($\epsilon = 0.3$) | 2% | 16% | **63%** | 30% | **42%** |
| $\ell_2$ PGD ($\epsilon = 2.0$) | 10% | 20% | **68%** | 55% | **59%** |
| $\ell_1$ PGD ($\epsilon = 10.0$) | 12% | 35% | **68%** | 15% | **48%** |
| Average | 8% | 23.67% | **66.33%** | 33.33% | **49.66%** |

### B.7 QUALITATIVE RESULTS

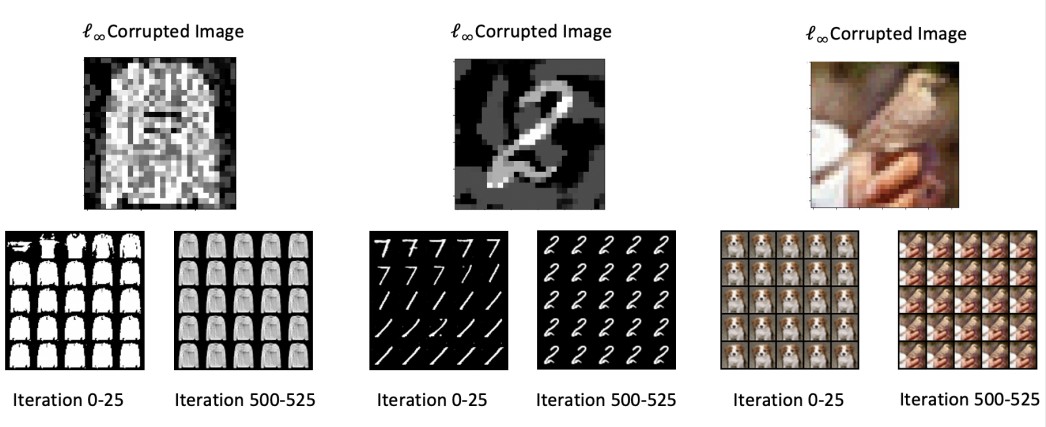

Figure 4: Qualitative Results for $\ell_\infty$ corrupted images for the MNIST, Fashion-MNIST, and CIFAR-10 datasets. Each $5 \times 5$ grid shows 25 iterations for $G(z^k)$ including the Defense-GAN initialization.

To qualitatively get a sense of whether GAN inversion succeeds at recovering the true image, we plot $G(z^k)$ as a function of $k$ for the different datasets. We focus on $\ell_\infty$ attacks although we note that the denoised results look qualitatively identical for the different attacks (while the corrupted image looks different). If the GAN inversion succeeds at denoising and modelling the clean data, then we

expect better attack detection accuracy since the $D_a c_a$ term can better capture the structure of the attack with limited noise. Figure 4 shows the results on the 3 datasets: MNIST, Fashion-MNIST, and CIFAR-10. Each grid of images shows 25 images corresponding to $G(z^k)$ for 25 iterations. Note that the iteration number includes the number of iterations needed for Defense-GAN initialization. The Defense-GAN initialization looks qualitatively similar to a clean image, but the iterations after alternating between updating $z$ and $c_a$ allow us to further classify the attack. In all 3 examples, we see successful inversion of the image despite starting from an incorrect class, further supporting the benign optimization landscape of the inversion problem.

### B.8 Using Diffusion Models Instead of GANs

With the development of diffusion models that have been shown to be powerful generative models, a natural question is whether our approach can be extended to use diffusion models as the underlying generative model in practice. We emphasize that the main contribution of our work is a theoretically grounded algorithm for reverse engineering deceptions, and using diffusion models presents a vast set of challenges in providing recovery guarantees for the underlying signal and attack.

Nie et al. (2022) provide a method for adversarial purification using diffusion models, which they call DiffPure, with the hope that by passing an adversarially corrupted image to the forward and backward process of a diffusion model, adversarial noise is removed. On its own, DiffPure does not provide a way of joint signal and attack classification. However, using our approach, one can alternate between denoising using diffusion models and classifying the attack using the attack dictionary as we have done in this work. Specifically, let $D(x')$ be the denoised version of $x'$ by passing through the forward and reverse process of a diffusion model $D$. Then, our modelling assumption is that $x' \approx D(x') + D_a c_a$. Our alternating algorithm then becomes in each step to perform two operations: a) compute $D(x')$ b) use proximal gradient descent steps to fit $\|x' - D(x') - D_a c_a\|_2 + \lambda_2 \|c_a\|_{1,2}$ for $c_a$. We call this method BSD-DP and BSD-DP-AD for signal and attack classification respectively (where DP stands for DiffPure). The below table shows results of this method on the CIFAR-10 dataset.

We observe a few interesting phenomena when using diffusion models. First, the DiffPure signal classifier removes the adversarial noise well, which hints to the success of the DiffPure approach as observed by Nie et al. (2022). When, we add the modelling of the attack however, we observe slightly improved signal classification results with the intuition that part of the adversarial noise is also captured by the $D_a c_a$ model. However, as evidenced by the BSD-DP-AD column, the attack model is insufficient at classifying the correct attack type surprisingly predicting that all attacks are of type $\ell_2$. We believe the reason for this is that the diffusion model as an underlying generative model gives very good results but on average, $D(x')$ is not close to $x'$ as an exact reconstruction. Indeed, in our experiments, for $\ell_2$ attacks we find that on average, $\|D(x') - x'\|_2 \approx 10$ (which is much higher than the strength of the $\ell_2$ perturbation) for the 100 test examples shown here. In contrast, the GAN inversion produces almost exact inversion of the underlying image, which is more beneficial for our model. If we use fewer forward steps in the diffusion model, we might expect to recover the underlying signal better, but not remove the adversarial component. We defer a more thorough evaluation of this tradeoff to future work. Despite this, it is surprising that the signal classification still improves by modelling the attack structure using the dictionary. These experiments showcase the benefit of using GANs to model the underlying distribution since provably near-exact inversion helps the modelling of the clean data and gives on average higher attack classification for only slightly reduced signal classification accuracy.

Table 6: Adversarial image and attack classification accuracy on CIFAR-10 dataset for 100 test examples. See text above for column descriptions. DP stands for DiffPure.

| **CIFAR-10** | CNN | BSD-GAN | DP | BSD-DP | BSD-GAN-AD | BSD-DP-AD |
|:---:|:---:|:---:|:---:|:---:|:---:|:---:|
| Clean accuracy | 99% | 72% | 84% | 84% | - | - |
| $\ell_\infty$ PGD ($\epsilon = 0.03$) | 0% | 76% | 82% | **87%** | **48%** | 0% |
| $\ell_2$ PGD ($\epsilon = 0.5$) | 0% | **87%** | 83% | **87%** | 77% | **100%** |
| $\ell_1$ PGD ($\epsilon = 12.0$) | 0% | 71% | 82% | **87%** | 44% | 0% |
| Average | 0% | 78% | 82.33% | **87%** | **56%** | 33.33% |

