# OpenReview forum: "A Linearly Convergent GAN Inversion-based Algorithm for Reverse Engineering of Deceptions"
_ICLR.cc/2024/Conference — Submitted to ICLR 2024_

### Official Review · Reviewer_sjuu · 2023-11-01

**Soundness:** 2 fair
**Presentation:** 2 fair
**Contribution:** 2 fair
**Rating:** 3
**Confidence:** 2

**Summary:**

In this paper, the authors present an approach to reverse engineering adversarial attacks. They leverage the generative priors of Generative Adversarial Networks (GANs) and utilize block-sparse representations in attack dictionaries. This approach offers deterministic linear convergence guarantees for the problem. The authors provide results on MNIST, Fashion-MNIST, and CIFAR-10.

**Strengths:**

This paper works on Reverse Engineering of Deceptions (RED) problem. It tries to defend against attacks and figure out how those attacks work. The RED problem is a new and practical area of research with high importance.

**Weaknesses:**

1. One important contribution of this paper is that it has theoretical proof. However, the proof is based on the assumption that the activation functions in the described network are smooth and twice differentiable. However, it is not a very practical assumption.

2. From the dataset aspect, the paper only shows results on simple datasets. There is no result validating the performance on a more complicated and widely adopted dataset such as ImageNet. The comparison baselines are also limited. The authors only compare with SBSAD. The authors mentioned other RED methods without theoretical guarantees such as [10] [11] [25], but not comparisons are included. Furthermore, the paper only have results on PGD attack. It's not sure if the method can be generalized to other more recent attack types such as AutoAttack. It's also meaning to consider the adaptive attack case.

**Questions:**

1. Can this method generalize to more complicated dataset such as ImageNet?
2. How is the recovered clean signal look like? Is it close to the groundtruth clean example?
3. What will the performance be like if the attack is not PGD, such as AutoAttack and Feature attack?

---

> ### Author Response · Authors · 2023-11-21
>
> We would like to highlight our main concern with this review. We noticed that this review is copy-pasted directly from a private review of this paper at another venue. We will respond to each comment individually below but note that several of these comments are addressed in the main paper already, so we find this review to be an unfair evaluation of our paper.
>
> > “One important contribution of this paper is that it has theoretical proof. However, the proof is based on the assumption that the activation functions in the described network are smooth and twice differentiable. However, it is not a very practical assumption.”
>
> - First, there appears to be a confusion between the role of the assumption as a means to prove theoretical results and the practicality of the method. Our theoretical results demonstrate linear convergence guarantees under our set of assumptions. We note that the only prior works on this problem (as cited in our paper), provide convergence guarantees in the setting where the weights of the GAN are random or close to random. In contrast, our work is the first to move beyond this set of assumptions and provide a different convergence analysis. While in many cases assumptions can be too strong, one can often run the associated algorithm with or without these assumptions. In other words, theoretical assumptions need not impact the practicality of the method. Moreover, in this particular case, our assumption is weaker than existing assumptions made by relevant state-of-the-art work.
>
> - The “Remark on Assumptions” in Section 5.2 and Appendix Section B.4 explicitly comment on the role of Assumption 1 on practical experiments and demonstrate the assumption does not affect inversion in practice. Thus, we argue our method has both theoretical and practical merits.
>
>
>  > “From the dataset aspect, the paper only shows results on simple datasets. There is no result validating the performance on a more complicated and widely adopted dataset such as ImageNet. The comparison baselines are also limited. The authors only compare with SBSAD. The authors mentioned other RED methods without theoretical guarantees such as [10] [11] [25], but not comparisons are included. Furthermore, the paper only have results on PGD attack. It's not sure if the method can be generalized to other more recent attack types such as AutoAttack. It's also meaning to consider the adaptive attack case.
>
>
>
> - We understand the concern the reviewer has on the datasets chosen for this paper. We would like to emphasize that our work is the first one to propose not only a nonlinear framework for this problem, but also a method with theoretical guarantees (as compared to existing black-box methods for related RED problems). Even the three nonlinear evaluation datasets in the paper highlight that prior state-of-the-art attack classification results with theoretical guarantees from Thaker et al fail at modelling the clean signal. Further, the key strength of our model is that any generative model can be used to model the clean data as long as it has favorable inversion properties. The StyleGAN-XL paper provides impressive empirical GAN inversion results on Imagenet, and it can easily be substituted as the generative model in our approach. For computational reasons, we could not perform comprehensive Imagenet extensions for the rebuttal period. However, we have some preliminary results on TinyImagenet, which show that for 100 test examples using $\ell_\infty$ and $\ell_2$PGD attacks separately on an AlexNet architecture, we have already observed a signal classification accuracy increase from 11% undefended to 39% using GAN inversion. We believe that the existing theoretical results and the current set of experiments nonetheless provide strong evidence for the merits of the proposed novel approach in reverse engineering attacks.
>
> - We are not sure what references 10, 11, 25 in this paper are as the references are by author name.
>
> - Our aim is provable attack classification using a dictionary model for the attacks. The work of Thaker et al provides a theoretical justification for using the dictionary model for PGD $\ell_p$ attacks (namely that attacks on test examples can be represented as linear combinations of attacks on training examples for a network that uses ReLU activations). It is not clear whether this model makes sense for more complex attacks such as AutoAttack, and so we defer a model for these classes of attacks to future work.
>
> > “How is the recovered clean signal look like? Is it close to the groundtruth clean example?”
>
> Appendix Section B.4 provides qualitative results for reconstruction error and we observe that the reconstructed image is indeed close to the ground truth denoised image.

---

> > ### Comment · Reviewer_sjuu · 2023-11-23
> >
> > Thank the authors for providing the response. The main contributions of the paper mentioned by the authors is that they propose a GAN inversion-based RED algorithm with linear convergence guarantees.  The assumption for the algorithm is still quite strong by assuming the activation function is twice differentiable and smooth for the GAN inversion problem. Most of the widely used GAN models such as Style-GAN does not satisfy this assumption. Though the authors try to show their method's performance when violating the assumptions by conducting experiments using Style-GAN-XL, it makes me confused of why it's worth exploring as most existing GANs does not satisfy the assumption. What is the meaning to show the convergence guarantees when it does not hold for most commonly used GANs? I don't see the benefits of this theoretical proof of the convergence guarantees giving this strong assumption for the understanding of the RED problem.
> >
> > It's good to see the authors providing the results on the Tiny-ImageNet dataset in the rebuttal, but I think the paper can be much more improved with experiments on large scale dataset such as ImageNet.
> >
> > The authors mentioned that their method is based on using a dictionary model for the attacks, which is for PGD $l_p$ attacks. While PGD attack is a widely used attack method, it represents only a subset of the possible attack methods. As mentioned in the introduction of the paper, the objective of RED is to defend against the attack and infer the deception strategy. Demonstrating results only on PGD attacks might not be comprehensive enough. When an adversarial example is presented, how to know if it's PGD attack to let this method work?
> >
> > Given the limitations, I maintain my current rating of this paper.

---

### Official Review · Reviewer_yzAo · 2023-11-02

**Soundness:** 2 fair
**Presentation:** 1 poor
**Contribution:** 2 fair
**Rating:** 5
**Confidence:** 3

**Summary:**

While previous research focused on defense strategies, recent work has explored reverse engineering the attack process to understand and classify threats. However, existing methods lack theoretical guarantees, as they assume data lies in linear subspaces, which may not hold for complex datasets. This paper introduces a framework that assumes clean data resides in the range of a GAN (Generative Adversarial Network). To classify signals and attacks, it jointly addresses a GAN inversion problem and a block-sparse recovery problem, offering deterministic linear convergence guarantees, a first in this context. Empirical results on nonlinear datasets show the effectiveness of this approach compared to existing methods.

**Strengths:**

1) Authors introduce a framework that combines GANs, which has connection with the clean data.

2) This paper conducts a thorough theoretical analysis to prove the effectiveness of the proposed method.

3) Authors meticulously designed experiments, thoroughly analyzed the experimental results, and demonstrated the effectiveness of the algorithm.

**Weaknesses:**

1） In the introduction section of this article, authors use too much space to introducing the background,  and takes too long to present the problem. The article lacks effective organization.
2) The presentation of this article needs improvement, as there are many grammar issues, and it lacks readability. For example,  in the third sentence of second paragraph, citing the related works between two commas.
3) The experiment is not much convincing, since it uses small datasets.
4）The results of the experiment cannot strongly support the algorithm's advantages.

**Questions:**

My mainly concern is with the writing of the article. The sentences are not easy to understand, there are grammar errors, and the readability is poor.

---

> ### Author Response · Authors · 2023-11-21
>
> We thank the reviewer for their feedback and their appreciation of our thorough theoretical analysis, meticulous design of experiments, and thorough analysis of results. Below, we address each of the comments individually.
>
> > “In the introduction section of this article, authors use too much space to introducing the background, and takes too long to present the problem. The article lacks effective organization.”
>
>
>
> Thank you for the feedback. We have condensed the introduction slightly to motivate the problem setup of RED more directly and uploaded a revision. We felt it was important to motivate the problem setup of RED in this way since it is distinctly different from standard literature in adversarial learning. Further, our approach builds closely on the prior work of Thaker et al, so we believe the discussion of the flaws in this work is crucial to understanding our contributions. We hope the reviewer finds that the current modifications to the introduction address their comments, but we would appreciate if the reviewer has more specific feedback on the presentation. Moreover, the rest of the paper is structured in a didactic manner: starting from problem setup, moving to a theoretical analysis structured in 3 settings that go in order of added complexity, and concluding with a discussion of experimental results. We respectfully disagree that the article lacks effective organization, but we would appreciate any specific feedback the reviewer has on improving readability and organization.
>
> > “The presentation of this article needs improvement, as there are many grammar issues, and it lacks readability. For example, in the third sentence of second paragraph, citing the related works between two commas.”
>
> We respectfully disagree with the reviewer that there are many grammar issues with the article. Citing related works between two commas seems like a style choice as opposed to a grammar issue. To be consistent though, we have removed the extra comma in the three other citations in the paper which had that issue. Upon a thorough proofread, we have found no other grammar issues. We would appreciate if the reviewer has specific suggestions to further improve readability.
>
>
>
> > “The experiment is not much convincing, since it uses small datasets. 4）The results of the experiment cannot strongly support the algorithm's advantages.”
>
>
>
> We attempted to perform experiments that convincingly showcase our theoretical results and the merits of the proposed algorithm, and we are glad that the reviewer found that we “meticulously designed experiments, thoroughly analyzed the experimental results, and demonstrated the effectiveness of the algorithm.” We understand the concern the reviewer has on the datasets chosen for this paper. We would like to emphasize that our work is the first one to propose not only a nonlinear framework for this problem, but also a method with theoretical guarantees (as compared to existing black-box methods for related RED problems). Even the three nonlinear evaluation datasets in the paper highlight that prior state-of-the-art attack classification results with theoretical guarantees from Thaker et al fail at modelling the clean signal. Further, the key strength of our model is that any generative model can be used to model the clean data as long as it has favorable inversion properties. The StyleGAN-XL paper provides impressive empirical GAN inversion results on Imagenet, and it can easily be substituted as the generative model in our approach. For computational reasons, we could not perform comprehensive Imagenet extensions for the rebuttal period. However, we have some preliminary results on TinyImagenet, which show that for 100 test examples using $\ell_\infty$ and $\ell_2$ PGD attacks separately on an AlexNet architecture, we have already observed a signal classification accuracy increase from 11% undefended to 39% using GAN inversion. We believe that the existing theoretical results and the current set of experiments nonetheless provide strong evidence for the merits of the proposed novel approach in reverse engineering attacks.

---

### Official Review · Reviewer_MrgY · 2023-11-09

**Soundness:** 2 fair
**Presentation:** 3 good
**Contribution:** 2 fair
**Rating:** 6
**Confidence:** 3

**Summary:**

This paper introduces a GAN inversion-based approach to reverse engineering of deceptions (RED), providing provable guarantees.  The objective of RED extends beyond mere defense against attacks; it encompasses the ability to reverse engineer and deduce the specific nature of the attack. The approach deviates from previous assumptions that clean data is contained within linear subspaces, instead utilizing the capabilities of nonlinear deep generative models. The research paper presents a theoretical analysis that demonstrates the achievement of linear convergence towards global optima in the context of the nonconvex inverse problem, subject to certain local error bound conditions. The proposed model's robustness is demonstrated through empirical validations conducted on the MNIST, Fashion-MNIST, and CIFAR-10 datasets.

**Strengths:**

+ Contrary to adopting overly constraining assumptions like assuming data resides in a union of linear subspaces, which proves inadequate for intricate datasets, or employing networks with randomized weights, as some studies have done, this research opts for more rational assumptions. Specifically, it embraces considerations such as the Local Error Bound Condition (highlighted as Assumptions 2 and 5 in the paper) and proximal Polyak-Łojasiewicz conditions.
+ This study has not only addressed the theoretical aspect but has also presented empirical proof across various datasets.
+ In addition to conducting experiments with real data, the researchers also performed experiments using synthetic data, thereby contributing further evidence to their study.

**Weaknesses:**

- The paper could enhance its comparison with State-of-the-Art methods, particularly by evaluating signal accuracy in comparison to other leading works. A broader array of these methods should be taken into account for a more comprehensive comparison.
- The evaluation included a comparison of performance in terms of adversarial signal classification accuracy and attack classification accuracy. In addition to assessing robust accuracy, it would be beneficial to extend the comparison to include clean accuracy.
- The assessment could extend to more sophisticated datasets, like ImageNet, as numerous other State-of-the-Art methods have previously reported their performance on such datasets.
- To empirically demonstrate the Error Bound, it would be advantageous to present it across a set of samples with varying sizes and illustrate its consistency. If the consistency holds, the maximum/standard deviation should remain nearly constant with an increase in the number of data samples.
- The effectiveness of GAN inversion falls short compared to the performance achieved with diffusion models, which are commonly employed in current practical applications. Some studies, such as [Nie et al.](https://arxiv.org/abs/2205.07460), have explored the use of Diffusion Models for purification, resulting in enhanced performance. It might be worthwhile to incorporate these models in this context, considering their potential for estimating clean data. Additionally, comparing the results against such approaches could provide valuable insights.

**Questions:**

1) Why was the testing limited to only 100 test examples for the CIFAR-10 dataset?
2) In practical applications, subgradient descent is typically not employed, as was done in your optimization for GAN inversion. Have you compared your GAN inversion results against those obtained using alternative methods?

---

> ### Author Response · Authors · 2023-11-21
>
> We thank the reviewer for their feedback and appreciation of the importance of the RED problem, the theoretical results that moves beyond the overly constraining assumptions of prior work, and empirical results on both synthetic and real data! Below, we address each of the comments individually.
>
> > “The paper could enhance its comparison with State-of-the-Art methods, particularly by evaluating signal accuracy in comparison to other leading works. A broader array of these methods should be taken into account for a more comprehensive comparison. The effectiveness of GAN inversion falls short compared to the performance achieved with diffusion models, which are commonly employed in current practical applications. Some studies, such as Nie et al., have explored the use of Diffusion Models for purification, resulting in enhanced performance. It might be worthwhile to incorporate these models in this context, considering their potential for estimating clean data. Additionally, comparing the results against such approaches could provide valuable insights.”
>
> Thank you for the feedback. In the revised Appendix of the paper, we have added section B.8 and Table 6 which provide a comparison of our approach and the method that uses diffusion models for the underlying generative model instead of GANs. We show the table here as well. Using our approach, one can alternate between denoising using diffusion models and classifying the attack using the attack dictionary as we have done in this work. Specifically, let $D(x')$ be the denoised version of $x'$ by passing through the forward and reverse process of a diffusion model $D$ (DiffPure approach from Nie et al.) . Then, our modelling assumption is that $x' \approx D(x') + D_a c_a$. Our alternating algorithm then becomes in each step to perform two operations: a)  compute $D(x')$ b) use proximal gradient descent steps to fit the model for $c_a$. We call this method BSD-DP and BSD-DP-AD for signal and attack classification respectively (where DP stands for DiffPure).
>
> We find that while the diffusion model is a powerful denoiser and yields slightly improved signal classification accuracy on average over different $\ell_p$ attacks, it does not result in accurate reconstructions of the original image, which yields poor attack classification accuracy compared to our approach using GANs. Please refer to Appendix Section B.8 for more analysis. Further, a strong benefit of our approach is that it comes with theoretical guarantees on GAN inversion and attack classification, whereas provable denoising using diffusion models with complex nonlinear reverse processes is an open problem.
>
> ### Adversarial image and attack classification accuracy on CIFAR-10 dataset for 100 test examples. See text above for column descriptions. DP stands for DiffPure.
>
> | CIFAR-10 | CNN | BSD-GAN | DP | BSD-DP | BSD-GAN-AD | BSD-DP-AD |
> |----------|-----|---------|----|--------|------------|------------|
> | Clean accuracy | 99% | 72% | 84% | 84% | - | - |
> | $\ell_\infty$ PGD ($\epsilon = 0.03$) | 0% | 76% | 82% | **87%** | **48%** | 0% |
> | $\ell_2$ PGD ($\epsilon = 0.5$) | 0% | **87%** | 83% | **87%** | 77% | **100%** |
> | $\ell_1$ PGD ($\epsilon = 12.0$) | 0% | 71% | 82% | **87%** | 44% | 0% |
> | Average | 0% | 78% | 82.33% | **87%** | **56%** | 33.33% |
>
> > “The evaluation included a comparison of performance in terms of adversarial signal classification accuracy and attack classification accuracy. In addition to assessing robust accuracy, it would be beneficial to extend the comparison to include clean accuracy.”
>
> Thank you for pointing this out, Table 1 already contains a row for clean accuracy on the MNIST dataset, but we have modified the paper to include clean accuracy on the CIFAR-10 dataset as well.

---

> > ### Author Response · Authors · 2023-11-21
> >
> > > “The assessment could extend to more sophisticated datasets, like ImageNet, as numerous other State-of-the-Art methods have previously reported their performance on such datasets.
> >
> > We understand the concern the reviewer has on the datasets chosen for this paper. We would like to emphasize that our work is the first one to propose not only a nonlinear framework for this problem, but also a method with theoretical guarantees (as compared to existing black-box methods for related RED problems). Even the three nonlinear evaluation datasets in the paper highlight that prior state-of-the-art attack classification results with theoretical guarantees from Thaker et al fail at modelling the clean signal. Further, the key strength of our model is that any generative model can be used to model the clean data as long as it has favorable inversion properties. The StyleGAN-XL paper provides impressive empirical GAN inversion results on Imagenet, and it can easily be substituted as the generative model in our approach. For computational reasons, we could not perform comprehensive Imagenet extensions for the rebuttal period. However, we have some preliminary results on TinyImagenet, which show that for 100 test examples using $\ell_\infty$ and $\ell_2$ PGD attacks separately on an AlexNet architecture, we have already observed a signal classification accuracy increase from 11% undefended to 39% using GAN inversion. We believe that the existing theoretical results and the current set of experiments nonetheless provide strong evidence for the merits of the proposed novel approach in reverse engineering attacks.
> >
> > > “To empirically demonstrate the Error Bound, it would be advantageous to present it across a set of samples with varying sizes and illustrate its consistency. If the consistency holds, the maximum/standard deviation should remain nearly constant with an increase in the number of data samples.”
> >
> >
> > While we are not exactly sure what the reviewer means by consistency here, we interpret it as varying the number of samples over which the local error bound parameter is calculated to see if the trend holds more broadly. Note that the GAN inversion is ran as a different optimization problem per datapoint, and the landscape shown in Figure 2 holds for all choices of $z^*$ (as Corollary 4 argues), even though only one such figure is shown. For deep networks with random weights, following the setup in Appendix Section B.3, we have rerun the local error bound calculation with a varying number of datapoints over which the average $\mu$ is computed. We found that the average and standard deviation is roughly constant in the increase of data samples as expected. We thank you for the suggestion and hope this is a satisfactory answer to your point.
> >
> > ### Consistency of Local Error Bound Condition as Number of Samples Increases
> >  | # Examples | 10     | 25     | 50     | 75     | 100    | 150    | 200    |
> > |------------|--------|--------|--------|--------|--------|--------|--------|
> > | Average    | 470.74 | 487.99 | 434.32 | 502.43 | 508.65 | 468.71 | 447.82 |
> > | Standard deviation | 268.45 | 218.97 | 192.40 | 199.34 | 254.58 | 236.02 | 310.11 |
> >
> >
> > > “Why was the testing limited to only 100 test examples for the CIFAR-10 dataset?”
> >
> > Thank you for the comment, we have observed similar performance when running GAN inversion for 500 test examples as well. For example, we can still achieve 74%, 71.8% and 69.8% signal classification accuracy for $\ell_1$, $\ell_2$ and $\ell_\infty$ PGD attacks.
> >
> > > “In practical applications, subgradient descent is typically not employed, as was done in your optimization for GAN inversion. Have you compared your GAN inversion results against those obtained using alternative methods?”
> >
> > We designed most of our experiments to be in the regime that our theoretical results were applicable in and comparable to existing theoretical results in GAN inversion such as the work of Hand et al. As our convergence theory is proven for subgradient descent, we chose that algorithm for all our experiments. However, even in practical applications, we have seen inversion using subgradient descent (for example from StyleGAN-XL paper) but in the style space of the GAN, which is what we adopted for our StyleGAN experiments as well. We note though that the proposed framework is general and any GAN inversion algorithm could be used in conjunction with proximal gradient descent.

---

> > > ### Comment · Reviewer_MrgY · 2023-11-22
> > >
> > > I would like to express my gratitude to the reviewers for their comprehensive feedback addressing each of my inquiries and areas of critique. The modifications made by the authors to the paper are duly acknowledged.
> > >
> > > I would also like to propose that the computational resources used for the empirical experiments be incorporated in the appendix as well.

---

> > > > ### Author Response · Authors · 2023-11-22
> > > >
> > > > Thank you for the positive feedback, and we are glad we were able to address all your concerns and critiques. We have added a subsection in Appendix Section B.2 about computational resources used. We would appreciate if the reviewer would consider increasing their score if we have addressed all their concerns.

---

### Meta-Review · Area_Chair_4JNo · 2023-12-07

**Metareview:**

The paper presents a novel approach to reverse engineering attacks using a nonlinear model that relies on inverting deep generative models, with a focus on GAN inversion and the application of theoretical guarantees. The reviewers raised concerns about the paper's experimental scope, particularly the lack of comprehensive comparison with state-of-the-art methods, limited exploration of datasets (including the absence of results on ImageNet), and the practical applicability of the theoretical assumptions made. The authors have addressed some of these concerns in their rebuttal, but significant issues remain, particularly regarding the breadth of datasets used for validation and the relevance of the theoretical assumptions to practical scenarios.

**Justification For Why Not Higher Score:**

A higher score is not justified due to the paper's limited experimental validation and the potential disconnect between its theoretical assumptions and practical applicability. The reviewers noted the lack of comparison with a broader array of state-of-the-art methods and the absence of results on more sophisticated datasets like ImageNet, which limits the paper's ability to demonstrate its effectiveness in diverse and complex scenarios.

**Justification For Why Not Lower Score:**

NA

---

### Decision · Program_Chairs · 2024-01-16

Reject